# Controlling Confusion via Generalisation Bounds

## Abstract

We establish new generalisation bounds for multiclass classification by abstracting to a more general setting of discretised error types. Extending the PAC-Bayes theory, we are hence able to provide fine-grained bounds on performance for multiclass classification, as well as applications to other learning problems including discretisation of regression losses. Tractable training objectives are derived from the bounds. The bounds are uniform over all weightings of the discretised error types and thus can be used to bound weightings not foreseen at training, including the full confusion matrix in the multiclass classification case.

## 1 Introduction

Generalisation bounds are a core component of the theoretical understanding of machine learning algorithms. For over two decades now, the PAC-Bayesian theory has been at the core of studies on generalisation abilities of machine learning algorithms. PAC-Bayes originates in the seminal work of [24, 25] and was further developed by citepcatoni2003pac,catoni2004statistical,catoni2007, among other authors—we refer to the recent surveys [16] and [1] for an introduction to the field. The outstanding empirical successes of deep neural networks in the past decade call for better theoretical understanding of deep learning, and PAC-Bayes emerged as one of the few frameworks allowing the derivation of meaningful (and non-vacuous) generalisation bounds for neural networks: the pioneering work of [13] has been followed by a number of contributions, including [28], [35], [19], [30, 31] and [4, 6, 5], to name but a few.

Much of the PAC-Bayes literature focuses on the case of binary classification, or of multiclass classification where one only distinguishes whether each classification is correct or incorrect. This is in stark contrast to the complexity of contemporary real-world learning problems. This work aims to bridge this gap via generalisation bounds that provide information rich measures of performance at test time by controlling the probabilities of errors of any finite number of types, bounding combinations of these probabilities uniformly over all weightings.

**Previous results.** We believe our framework of discretised error types to be novel. In the particular case of multiclass classification, little is known from a theoretical perspective and, to the best of our knowledge, only a handful of relevant strategies or generalisation bounds can be compared to the present paper. The closest is the work of [27] on a PAC-Bayes generalisation bound on the operator norm of the confusion matrix, to train a Gibbs classifier. We focus on a different performance metric, in the broader setting of discretised error types. [17] suggest to minimise the confusion matrix norm with a focus on the imbalance between classes; their treatment is not done through PAC-Bayes. [18] extend the celebrated $\mathcal{C}$-bound in PAC-Bayes to weighted majority votes of classifiers, to perform multiclass classification. [3] present a streamlined version of some of the results from [27] in the case where some examples are voluntarily not classified (*e.g.*, in the case of too large uncertainty). More recently, [15] derive bounds for a majority vote classifier where the confusion matrix serves as an error indicator: they conduct a study of the Bayes classifier.

Submitted to 36th Conference on Neural Information Processing Systems (NeurIPS 2022). Do not distribute.

**From binary to multiclass classification.** A number of PAC-Bayesian bounds have been unified by a single general bound, found in [7]. Stated as Theorem 1 below, it applies to binary classification. We use it as a basis to prove our Theorem 3, a more general bound that can be applied to, amongst other things, multiclass classification and discretised regression. While the proof of Theorem 3 follows similar lines to that given in [7], our generalisation to 'soft' hypotheses incurring any finite number of error types requires a non-trivial extension of a result found in [22]. This extension (Lemma 5), along with its corollary (Corollary 6) may be of independent interest. The generalisation bound in [22], stated below as Corollary 2, is shown in [7] to be a corollary of their bound. In a similar manner, we derive Corollary 7 from Theorem 3. Obtaining this corollary is significantly more involved than the analogous derivation in [7] or the original proof in [22], requiring a number of technical results found in Appendix B.

Briefly, the results in [7] and [22] consider an arbitrary input set $\mathcal{X}$, output set $\mathcal{Y} = \{-1, 1\}$, hypothesis space $\mathcal{H} \subseteq \mathcal{Y}^{\mathcal{X}}$ and i.i.d. sample $S \in (\mathcal{X} \times \mathcal{Y})^m$. They then establish high probability bounds on the discrepancy between the risk (probability of error an a new datapoint) of any stochastic classifier $Q$ (namely, a distribution on $\mathcal{H}$) and its empirical counterpart (the fraction of the sample $Q$ misclassifies). The bounds hold uniformly over all $Q$ and contain a complexity term involving the Kullback-Leibler (KL) divergence between $Q$ and a reference distribution $P$ on $\mathcal{H}$ (often referred to as a prior by analogy with Bayesian inference—see the discussion in 16).

There are two ways in which the results in [7] and [22] can be described as binary. First, as $\mathcal{Y}$ contains two elements, this is obviously an instance of binary classification. But a more interesting and subtle way to look at this is that only two cases are distinguished—correct classification and incorrect classification. Specifically, since the two different directions in which misclassification can be made are counted together, the bound gives no information on which direction is more likely.

More generally, the aforementioned bounds can be applied in the context of multiclass classification provided one maintains the second binary characteristic by only distinguishing correct and incorrect classifications rather than considering the entire confusion matrix. However, note that these bounds will not give information on the relative likelihood of the different errors. In contrast, our new results can consider the entire confusion matrix, bounding how far the true (read "expected over the data-generating distribution") confusion matrix differs from the empirical one, according to some metric. In fact, our results extend to the case of arbitrary label set $\mathcal{Y}$, provided the number of different errors one distinguishes is finite.

Formally, we let $\bigcup_{j=1}^{M} E_j$ be a user-specified disjoint partition of $\mathcal{Y}^2$ into a finite number of $M$ *error types*, where we say that a hypothesis $h \in \mathcal{H}$ makes an error of type $j$ on datapoint $(x, y)$ if $(h(x), y) \in E_j$ (by convention, every pair $(\hat{y}, y) \in \mathcal{Y}^2$ is interpreted as a predicted value $\hat{y}$ followed by a true value $y$, in that order). It should be stressed that some $E_j$ need not correspond to mislabellings—indeed, some of the $E_j$ may distinguish different correct labellings. We then count up the number of errors of each type that a hypothesis makes on a sample, and bound how far this empirical distribution of errors is from the expected distribution under the data-generating distribution (Theorem 3). Thus, in our generalisation, the (scalar) risk and empirical risk ($R_D(Q)$ and $R_S(Q)$, defined in the next section) are replaced by $M$-dimensional vectors ($\boldsymbol{R}_D(Q)$ and $\boldsymbol{R}_S(Q)$), and our discrepancy measure $d$ is a divergence between discrete distributions on $M$ elements. Our generalisation therefore allows us to bound how far the true distribution of errors can be from the observed distribution of errors. If we then associate a loss value $\ell_j \in [0, \infty)$ to each $E_j$ we can derive a bound on the *total risk*, defined as the sum of the true error probabilities weighted by the loss values. In fact, the total risk is bounded with high probability uniformly over all such weightings. The loss values need not be distinct; we may wish to understand the distribution of error types even across error types that incur the same loss.

For example, in the case of binary classification with $\mathcal{Y} = \{-1, 1\}$, we can take the usual partition into $E_1 = \{(-1, -1), (1, 1)\}$ and $E_2 = \{(-1, 1), (1, -1)\}$ and loss values $\ell_1 = 0, \ell_2 = 1$, or the fine-grained partition $\mathcal{Y}^2 = \{(0, 0)\} \cup \{(1, 1)\} \cup \{(0, 1)\} \cup \{(1, 0)\}$ and the loss values $\ell_1 = \ell_2 = 0, \ell_3 = 1, \ell_4 = 2$. More generally, for multiclass classification with $N$ classes and $\mathcal{Y} = [N]$, one may take the usual coarse partition into $E_1 = \{(\hat{y}, y) \in \mathcal{Y}^2 : \hat{y} = y\}$ and $E_2 = \{(\hat{y}, y) \in \mathcal{Y}^2 : \hat{y} \neq y\}$ (with $\ell_1 = 0$ and $\ell_2 = 1$), or the fully refined partition into $E_{i,j} = \{(i, j)\}$ for $i, j \in [N]$ (with correspondingly greater choice of the associated loss values), or something in-between. Note that we still refer to $E_j$ as an "error type" even if it contains elements that correspond to correct classification, namely if there exists $y \in \mathcal{Y}$ such that $(y, y) \in E_j$. As we will see later, a more fine-grained

94 partition will allow more error types to be distinguished and bounded, at the expense of a looser
95 bound. As a final example, for regression with $\mathcal{Y} = \mathbb{R}$, we may fix $M$ strictly increasing thresholds
96 $0 = \lambda_1 < \lambda_2 < \cdots < \lambda_M$ and partition $\mathcal{Y}^2$ into $E_j = \{(y_1, y_2) \in \mathcal{Y}^2 : \lambda_j \leq |y_1 - y_2| < \lambda_{j+1}\}$ for
97 $j \in [M-1]$, and $E_M = \{(y_1, y_2) \in \mathcal{Y}^2 : |y_1 - y_2| \geq \lambda_M\}$.

98 **Outline.** We set our notation in Section 2. In Section 3 we state and prove generalisation bounds in
99 the setting of discretised error types: this significantly expands the previously known results from [7]
100 by allowing for generic output sets $\mathcal{Y}$. Our main results are Theorem 3 and Corollary 7. To make
101 our findings profitable to the broader machine learning community we then discuss how these new
102 bounds can be turned into tractable training objectives in Section 4 (with a general recipe described
103 in greater detail in Appendix A). The paper closes with perspectives for follow-up work in Section 5
104 and we defer to Appendix B the proofs of technical results.

## 2 Notation

106 For any set $A$, let $\mathcal{M}(A)$ be the set of probability measures on $A$. For any $M \in \mathbb{Z}_{>0}$, define
107 $[M] := \{1, 2, \ldots, M\}$, the $M$-dimensional simplex $\triangle_M := \{\boldsymbol{u} \in [0,1]^M : u_1 + \cdots + u_M = 1\}$
108 and its interior $\triangle_M^{>0} := \triangle_M \cap (0,1)^M$. For $m, M \in \mathbb{Z}_{>0}$, define the integer counterparts $S_{m,M} :=$
109 $\{(k_1, \ldots, k_M) \in \mathbb{Z}_{\geq 0}^M : k_1 + \cdots + k_M = m\}$ and $S_{m,M}^{>0} := S_{m,M} \cap \mathbb{Z}_{>0}^M$. The set $S_{m,M}$ is the
110 domain of the multinomial distribution with parameters $m, M$ and some $\boldsymbol{r} \in \triangle_M$, which is denoted
111 $\mathrm{Mult}(m, M, \boldsymbol{r})$ and has probability mass function for $\boldsymbol{k} \in S_{m,M}$ given by

$$\mathrm{Mult}(\boldsymbol{k}; m, M, \boldsymbol{r}) := \begin{pmatrix} m \\ k_1 \; k_2 \; \cdots \; k_M \end{pmatrix} \prod_{j=1}^M r_j^{k_j}, \quad \text{where} \quad \begin{pmatrix} m \\ k_1 \; k_2 \; \cdots \; k_M \end{pmatrix} := \frac{m!}{\prod_{j=1}^M k_j!}.$$

112 For $\boldsymbol{q}, \boldsymbol{p} \in \triangle_M$, let $\mathrm{kl}(\boldsymbol{q}\|\boldsymbol{p})$ denote the KL-divergence of $\mathrm{Mult}(1, M, \boldsymbol{q})$ from $\mathrm{Mult}(1, M, \boldsymbol{p})$, namely
113 $\mathrm{kl}(\boldsymbol{q}\|\boldsymbol{p}) := \sum_{j=1}^M q_j \ln \frac{q_j}{p_j}$, with the convention that $0 \ln \frac{0}{x} = 0$ for $x \geq 0$ and $x \ln \frac{x}{0} = \infty$ for $x > 0$.
114 For $M = 2$ we abuse notation and abbreviate $\mathrm{kl}((q, 1-q)\|(p, 1-p))$ to $\mathrm{kl}(q\|p)$, which is then the
115 conventional definition of $\mathrm{kl}(\cdot\|\cdot) : [0,1]^2 \to [0, \infty]$ found in the PAC-Bayes literature [as in 33, for
116 example].

117 Let $\mathcal{X}$ and $\mathcal{Y}$ be arbitrary input (*e.g.*, feature) and output (*e.g.*, label) sets respectively. Let $\bigcup_{j=1}^M E_j$
118 be a partition of $\mathcal{Y}^2$ into a finite sequence of $M$ *error types*, and to each $E_j$ associate a loss value
119 $\ell_j \in [0, \infty)$. The only restriction we place on the loss values $\ell_j$ is that they are not all equal. This is
120 not a strong assumption, since if they were all equal then all hypotheses would incur equal loss and
121 there would be no learning problem: we are effectively ruling out trivial cases.

122 Let $\mathcal{H} \subseteq \mathcal{Y}^{\mathcal{X}}$ denote a hypothesis class, $D \in \mathcal{M}(\mathcal{X} \times \mathcal{Y})$ a data-generating distribution and
123 $S \sim D^m$ an i.i.d. sample of size $m$ drawn from $D$. For $h \in \mathcal{H}$ and $j \in [M]$ we define the
124 *empirical $j$-risk* and *true $j$-risk* of $h$ to be $R_S^j(h) := \frac{1}{m} \sum_{(x,y) \in S} \mathbb{1}[(h(x), y) \in E_j]$ and $R_D^j(h) :=$
125 $\mathbb{E}_{(x,y) \sim D}[\mathbb{1}[(h(x), y) \in E_j]]$, respectively, namely, the proportion of the sample $S$ on which $h$ makes
126 an error of type $E_j$ and the probability that $h$ makes an error of type $E_j$ on a new $(x, y) \sim D$.

127 More generally, suppose $\mathcal{H} \subseteq \mathcal{M}(\mathcal{Y})^{\mathcal{X}}$ is a class of *soft* hypotheses of the form $H : \mathcal{X} \to \mathcal{M}(\mathcal{Y})$,
128 where, for any $A \subseteq \mathcal{Y}$, $H(x)[A]$ is interpreted as the probability according to $H$ that the label of
129 $x$ is in $A$. It is worth stressing that a soft hypothesis is still deterministic since a prediction is not
130 drawn from the distribution it returns. We then define the *empirical $j$-risk* of $H$ to be $R_S^j(H) :=$
131 $\frac{1}{m} \sum_{(x,y) \in S} H(x)[\{\hat{y} \in \mathcal{Y} : (\hat{y}, y) \in E_j\}]$, namely the mean—over the elements $(x, y)$ of $S$—
132 probability mass $H$ assigns to predictions $\hat{y} \in \mathcal{Y}$ incurring an error of type $E_j$ when labelling each $x$.
133 Further, we define the *true $j$-risk* of $H$ to be $R_D^j(H) := \mathbb{E}_{(x,y) \sim D}[H(x)[\{\hat{y} \in \mathcal{Y} : (\hat{y}, y) \in E_j\}]]$,
134 namely the mean—over $(x, y) \sim D$—probability mass $H$ assigns to predictions $\hat{y} \in \mathcal{Y}$ incurring an
135 error of type $E_j$ when labelling each $x$. We will see in Section 4 that the more general hypothesis
136 class $\mathcal{H} \subseteq \mathcal{M}(\mathcal{Y})^{\mathcal{X}}$ is necessary for constructing a differentiable training objective.

137 To each ordinary hypothesis $h \in \mathcal{Y}^{\mathcal{X}}$ there corresponds a soft hypothesis $H \in \mathcal{M}(\mathcal{Y})^{\mathcal{X}}$ that, for each
138 $x \in \mathcal{X}$, returns a point mass on $h(x)$. In this case, it is straightforward to show that $R_S^j(h) = R_S^j(H)$
139 and $R_D^j(h) = R_D^j(H)$ for all $j \in [M]$, where we have used the corresponding definitions above for
140 ordinary and soft hypotheses. Since, in addition, our results hold identically for both ordinary and

soft hypotheses, we henceforth use the same notation $h$ for both ordinary and soft hypotheses and their associated values $R_S^j(h)$ and $R_D^j(h)$. It will always be clear from the context whether we are dealing with ordinary or soft hypotheses and thus which of the above definitions of the empirical and true $j$-risks is being used.

We define the *empirical risk* and *true risk* of a (ordinary or soft) hypothesis $h$ to be $\boldsymbol{R}_S(h) :=$ $(R_S^1(h), \ldots, R_S^M(h))$ and $\boldsymbol{R}_D(h) := (R_D^1(h), \ldots, R_D^M(h))$, respectively. It is straightforward to show that $\boldsymbol{R}_S(h)$ and $\boldsymbol{R}_D(h)$ are elements of $\triangle_M$. Since $S$ is drawn i.i.d. from $D$, the expectation of the empirical risk is equal to the true risk, namely $\mathbb{E}_S[R_S^j(h)] = R_D^j(h)$ for all $j$ and thus $\mathbb{E}_S[\boldsymbol{R}_S(h)] = \boldsymbol{R}_D(h)$. Finally, we generalise to stochastic hypotheses $Q \in \mathcal{M}(\mathcal{H})$, which predict by first drawing a deterministic hypothesis $h \sim Q$ and then predicting according to $h$, where a new $h$ is drawn for each prediction. Thus, we define the *empirical $j$-risk* and *true $j$-risk* of $Q$ to be the scalars $R_S^j(Q) := \mathbb{E}_{h \sim Q}[R_S^j(h)]$ and $R_D^j(Q) := \mathbb{E}_{h \sim Q}[R_D^j(h)]$, for $j \in [M]$, and simply the *empirical risk* and *true risk* of $Q$ to be the elements of $\triangle_M$ defined by $\boldsymbol{R}_S(Q) := \mathbb{E}_{h \sim Q}[\boldsymbol{R}_S(h)]$ and $\boldsymbol{R}_D(Q) := \mathbb{E}_{h \sim Q}[\boldsymbol{R}_D(h)]$. As before, since $S$ is i.i.d., we have (using Fubini this time) that $\mathbb{E}_S[\boldsymbol{R}_S(Q)] = \boldsymbol{R}_D(Q)$. Finally, given a loss vector $\boldsymbol{\ell} \in [0, \infty)^M$, we define the *total risk* of $Q$ by the scalar $R_D^T(Q) := \sum_{j=1}^M \ell_j R_D^j(Q)$. As is conventional in the PAC-Bayes literature, we refer to sample independent and dependent distributions on $\mathcal{M}(\mathcal{H})$ (*i.e.* stochastic hypotheses) as *priors* (denoted $P$) and *posteriors* (denoted $Q$) respectively, even if they are not related by Bayes' theorem.

# 3  Inspiration and Main Results

We first state the existing results in [7] and [22] that we will generalise from just two error types (correct and incorrect) to any finite number of error types. These results are stated in terms of the scalars $R_S(Q) := \frac{1}{m} \sum_{(x,y) \in S} \mathbb{1}[h(x) \neq y]$ and $R_D(Q) := \mathbb{E}_{(x,y) \sim D} \mathbb{1}[h(x) \neq y]$ and, as we demonstrate, correspond to the case $M = 2$ of our generalisations.

**Theorem 1.** *(7, Theorem 4) Let $\mathcal{X}$ be an arbitrary set and $\mathcal{Y} = \{-1, 1\}$. Let $D \in \mathcal{M}(\mathcal{X} \times \mathcal{Y})$ be a data-generating distribution and $\mathcal{H} \subseteq \mathcal{Y}^{\mathcal{X}}$ be a hypothesis class. For any prior $P \in \mathcal{M}(\mathcal{H})$, $\delta \in (0, 1]$, convex function $d : [0, 1]^2 \to \mathbb{R}$, sample size $m$ and $\beta \in (0, \infty)$, with probability at least $1 - \delta$ over the random draw $S \sim D^m$, we have that simultaneously for all posteriors $Q \in \mathcal{M}(\mathcal{H})$*

$$d\big(R_S(Q), R_D(Q)\big) \leq \frac{1}{\beta}\left[\mathrm{KL}(Q\|P) + \ln \frac{\mathcal{I}_d(m, \beta)}{\delta}\right],$$

*with $\mathcal{I}_d(m, \beta) := \sup_{r \in [0,1]}\left[\sum_{k=0}^m \mathrm{Bin}(k; m, r) \exp\left(\beta d\left(\frac{k}{m}, r\right)\right)\right]$, where $\mathrm{Bin}(k; m, r)$ is the binomial probability mass function $\mathrm{Bin}(k; m, r) := \binom{m}{k} r^k (1-r)^{m-k}$.*

Note the original statement in [7] is for a positive integer $m'$, but the proof trivially generalises to any $\beta \in (0, \infty)$. One of the bounds that Theorem 1 unifies—which we also generalise—is that of [33], later tightened in [22], which we now state. It can be recovered from Theorem 1 by setting $\beta = m$ and $d(q, p) = \mathrm{kl}(q\|p) := q \ln \frac{q}{p} + (1-q) \ln \frac{1-q}{1-p}$.

**Corollary 2.** *(22, Theorem 5) Let $\mathcal{X}$ be an arbitrary set and $\mathcal{Y} = \{-1, 1\}$. Let $D \in \mathcal{M}(\mathcal{X} \times \mathcal{Y})$ be a data-generating distribution and $\mathcal{H} \subseteq \mathcal{Y}^{\mathcal{X}}$ be a hypothesis class. For any prior $P \in \mathcal{M}(\mathcal{H})$, $\delta \in (0, 1]$ and sample size $m$, with probability at least $1 - \delta$ over the random draw $S \sim D^m$, we have that simultaneously for all posteriors $Q \in \mathcal{M}(\mathcal{H})$*

$$\mathrm{kl}\big(R_S(Q), R_D(Q)\big) \leq \frac{1}{m}\left[\mathrm{KL}(Q\|P) + \ln \frac{2\sqrt{m}}{\delta}\right].$$

We wish to bound the deviation of the empirical vector $\boldsymbol{R}_S(Q)$ from the unknown vector $\boldsymbol{R}_D(Q)$. Since in general the stochastic hypothesis $Q$ we learn will depend on the sample $S$, it is useful to obtain bounds on the deviation of $\boldsymbol{R}_S(Q)$ from $\boldsymbol{R}_D(Q)$ that are uniform over $Q$, just as in Theorem 1 and Corollary 2. In Theorem 1, the deviation $d(R_S(Q), R_D(Q))$ between the scalars $R_S(Q), R_D(Q) \in [0, 1]$ is measured by some convex function $d : [0, 1]^2 \to \mathbb{R}$. In our case, the deviation $d(\boldsymbol{R}_S(Q), \boldsymbol{R}_D(Q))$ between the vectors $\boldsymbol{R}_S(Q), \boldsymbol{R}_D(Q) \in \triangle_M$ is measured by some convex function $d : \triangle_M^2 \to \mathbb{R}$. In Section 3.2 we will derive Corollary 7 from Theorem 3 by selecting $\beta = m$ and $d(\boldsymbol{q}, \boldsymbol{p}) := \mathrm{kl}(\boldsymbol{q}\|\boldsymbol{p})$, analogous to how Corollary 2 is obtained from Theorem 1.

## 3.1 Statement and proof of the generalised bound

We now state and prove our generalisation of Theorem 1. The proof follows identical lines to that of Theorem 1 given in [7], but with additional non-trivial steps to account for the greater number of error types and the possibility of soft hypotheses.

**Theorem 3.** *Let $\mathcal{X}$ and $\mathcal{Y}$ be arbitrary sets and $\bigcup_{j=1}^{M} E_j$ be a disjoint partition of $\mathcal{Y}^2$. Let $D \in \mathcal{M}(\mathcal{X} \times \mathcal{Y})$ be a data-generating distribution and $\mathcal{H} \subseteq \mathcal{M}(\mathcal{Y})^{\mathcal{X}}$ be a hypothesis class. For any prior $P \in \mathcal{M}(\mathcal{H})$, $\delta \in (0,1]$, jointly convex function $d : \triangle_M^2 \to \mathbb{R}$, sample size $m$ and $\beta \in (0,\infty)$, with probability at least $1 - \delta$ over the random draw $S \sim D^m$, we have that simultaneously for all posteriors $Q \in \mathcal{M}(\mathcal{H})$*

$$d\big(\boldsymbol{R}_S(Q), \boldsymbol{R}_D(Q)\big) \leq \frac{1}{\beta}\left[\mathrm{KL}(Q\|P) + \ln \frac{\mathcal{I}_d(m,\beta)}{\delta}\right], \tag{1}$$

*where $\mathcal{I}_d(m,\beta) := \sup_{\boldsymbol{r} \in \triangle_M}\left[\sum_{\boldsymbol{k} \in S_{m,M}} \mathrm{Mult}(\boldsymbol{k}; m, M, \boldsymbol{r}) \exp\left(\beta d\left(\frac{\boldsymbol{k}}{m}, \boldsymbol{r}\right)\right)\right]$. Further, the bounds are unchanged if one restricts to an ordinary hypothesis class, namely if $\mathcal{H} \subseteq \mathcal{Y}^{\mathcal{X}}$.*

The proof begins on the following page after a discussion and some auxiliary results. One can derive multiple bounds from this theorem, all of which then hold simultaneously with probability at least $1 - \delta$. For example, one can derive bounds on the individual error probabilities $R_D^j(Q)$ or combinations thereof. It is this flexibility that allows Theorem 3 to provide far richer information on the performance of the posterior $Q$ on unseen data. For a more in depth discussion of how such bounds can be derived, including a recipe for transforming the bound into a differentiable training objective, see Section 4 and Appendix A.

To see that Theorem 3 is a generalisation of Theorem 1, note that we can recover it by setting $\mathcal{Y} = \{-1, 1\}$, $M = 2$, $E_1 = \{(-y, y) : y \in \mathcal{Y}\}$ and $E_2 = \{(y, y) : y \in \mathcal{Y}\}$. Then, for any convex function $d : [0,1]^2 \to \mathbb{R}$, apply Theorem 3 with the convex function $d' : \triangle_M^2 \to \mathbb{R}$ defined by $d'((u_1, u_2), (v_1, v_2)) := d(u_1, v_1)$ so that Theorem 3 bounds $d'\big(\boldsymbol{R}_S(Q), \boldsymbol{R}_D(Q)\big) = d\big(R_S^1(Q), R_D^1(Q)\big)$ which equals $d\big(R_S(Q), R_D(Q)\big)$ in the notation of Theorem 1. Further,

$$\sum_{\boldsymbol{k} \in S_{m,2}} \mathrm{Mult}(\boldsymbol{k}; m, 2, \boldsymbol{r}) \exp\left(\beta d'\left(\frac{\boldsymbol{k}}{m}, \boldsymbol{r}\right)\right) = \sum_{k=0}^{m} \mathrm{Bin}(k; m, r_1) \exp\left(\beta d\left(\frac{k}{m}, r_1\right)\right),$$

so that the supremum over $r_1 \in [0, 1]$ of the right hand side equals the supremum over $\boldsymbol{r} \in \triangle_2$ of the left hand side, which, when substituted into (1), yields the bound given in Theorem 1.

Our proof of Theorem 3 follows the lines of the proof of Theorem 1 in [7], making use of the change of measure inequality Lemma 4. However, a complication arises from the use of soft classifiers $h \in \mathcal{M}(\mathcal{Y})^{\mathcal{X}}$. A similar problem is dealt with in [22] when proving Corollary 2 by means of a Lemma permitting the replacement of $[0, 1]$-valued random variables by corresponding $\{0, 1\}$-valued random variables with the same mean. We use a generalisation of this, stated as Lemma 5 (Lemma 3 in 22 corresponds to the case $M = 2$), the proof of which is not insightful for our purposes and thus deferred to Appendix B.1. An immediate consequence of Lemma 5 is Corollary 6, which is a generalisation of the first half of Theorem 1 in [22]. While we only use it implicitly in the remainder of the paper, we state it as it may be of broader interest.

The consequence of Lemma 5 is that the worst case (in terms of bounding $d(\boldsymbol{R}_S(Q), \boldsymbol{R}_D(Q))$) occurs when $\boldsymbol{R}_{\{(x,y)\}}(h)$ is a one-hot vector for all $(x, y) \in S$ and $h \in \mathcal{H}$, namely when $\mathcal{H} \subseteq \mathcal{M}(\mathcal{Y})^{\mathcal{X}}$ only contains hypotheses that, when labelling $S$, put all their mass on elements $\hat{y} \in \mathcal{Y}$ that incur the same error type[1]. In particular, this is the case for hypotheses that put all their mass on a single element of $\mathcal{Y}$, equivalent to the simpler case $\mathcal{H} \subseteq \mathcal{Y}^{\mathcal{X}}$ as discussed in Section 2. Thus, Lemma 5 shows that the bound given in Theorem 3 cannot be made tighter only by restricting to such hypotheses.

**Lemma 4.** *(Change of measure, 10, 11) For any set $\mathcal{H}$, any $P, Q \in \mathcal{M}(\mathcal{H})$ and any measurable function $\phi : \mathcal{H} \to \mathbb{R}$, $\mathbb{E}_{h \sim Q} \phi(h) \leq \mathrm{KL}(Q\|P) + \ln \mathbb{E}_{h \sim P} \exp(\phi(h))$.*

**Lemma 5.** *(Generalisation of Lemma 3 in 22) Let $\boldsymbol{X}_1, \ldots, \boldsymbol{X}_m$ be i.i.d $\triangle_M$-valued random vectors with mean $\boldsymbol{\mu}$ and suppose that $f : \triangle_M^m \to \mathbb{R}$ is convex. If $\boldsymbol{X}_1', \ldots, \boldsymbol{X}_m'$ are i.i.d. $\mathrm{Mult}(1, M, \boldsymbol{\mu})$ random vectors, then $\mathbb{E}[f(\boldsymbol{X}_1, \ldots, \boldsymbol{X}_m)] \leq \mathbb{E}[f(\boldsymbol{X}_1', \ldots, \boldsymbol{X}_m')]$.*

---

[1]More precisely, when $\forall h \in \mathcal{H} \ \forall (x, y) \in S \ \exists j \in [M]$ such that $h(x)[\{\hat{y} \in \mathcal{Y} : (\hat{y}, y) \in E_j\}] = 1$.

**Corollary 6.** *(Generalisation of Theorem 1 in [22]) Let $X_1, \ldots, X_m$ be i.i.d $\triangle_M$-valued random vectors with mean $\boldsymbol{\mu}$, and $X'_1, \ldots, X'_m$ be i.i.d.* $\mathrm{Mult}(1, M, \boldsymbol{\mu})$. *Define $\bar{X} := \frac{1}{m} \sum_{i=1}^{m} X_i$ and $\bar{X}' := \frac{1}{m} \sum_{i=1}^{m} X'_i$. Then $\mathbb{E}[\exp(m\mathrm{kl}(\bar{X}\|\mu))] \le \mathbb{E}[\exp(m\mathrm{kl}(\bar{X}'\|\mu))]$.*

*Proof.* (of Corollary 6) This is immediate from Lemma 5 since the average is linear, the kl-divergence is convex and the exponential is non-decreasing and convex. □

*Proof.* (of Theorem 3) The case $\mathcal{H} \subseteq \mathcal{Y}^{\mathcal{X}}$ follows directly from the more general case by taking $\mathcal{H}' := \{h' \in \mathcal{M}(\mathcal{Y})^{\mathcal{X}} : \exists h \in \mathcal{H} \text{ such that } \forall x \in \mathcal{X} \ h'(x) = \delta_{h(x)}\}$, where $\delta_{h(x)} \in \mathcal{M}(\mathcal{Y})$ denotes a point mass on $h(x)$. For the general case $\mathcal{H} \subseteq \mathcal{M}(\mathcal{Y})^{\mathcal{X}}$, using Jensen's inequality with the convex function $d(\cdot, \cdot)$ and Lemma 4 with $\phi(h) = \beta d(\boldsymbol{R}_S(h), \boldsymbol{R}_D(h))$, we see that for all $Q \in \mathcal{M}(\mathcal{H})$

$$
\begin{aligned}
\beta d\big(\boldsymbol{R}_S(Q), \boldsymbol{R}_D(Q)\big) &= \beta d\left(\mathop{\mathbb{E}}_{h \sim Q} \boldsymbol{R}_S(h), \mathop{\mathbb{E}}_{h \sim Q} \boldsymbol{R}_D(h)\right) \\
&\le \mathop{\mathbb{E}}_{h \sim Q} \beta d\big(\boldsymbol{R}_S(h), \boldsymbol{R}_D(h)\big) \\
&\le \mathrm{KL}(Q\|P) + \ln\left(\mathop{\mathbb{E}}_{h \sim P} \exp\big(\beta d\big(\boldsymbol{R}_S(h), \boldsymbol{R}_D(h)\big)\big)\right) \\
&= \mathrm{KL}(Q\|P) + \ln(Z_P(S)),
\end{aligned}
$$

where $Z_P(S) := \mathbb{E}_{h \sim P} \exp\big(\beta d(\boldsymbol{R}_S(h), \boldsymbol{R}_D(h))\big)$. Note that $Z_P(S)$ is a non-negative random variable, so that by Markov's inequality $\mathop{\mathrm{P}}_{S \sim D^m}\left(Z_P(S) \le \frac{\mathbb{E}_{S' \sim D^m} Z_P(S')}{\delta}\right) \ge 1 - \delta$. Thus, since $\ln(\cdot)$ is strictly increasing, with probability at least $1 - \delta$ over $S \sim D^m$, we have that simultaneously for all $Q \in \mathcal{M}(\mathcal{H})$

$$
\beta d\big(\boldsymbol{R}_S(Q), \boldsymbol{R}_D(Q)\big) \le \mathrm{KL}(Q\|P) + \ln \frac{\mathop{\mathbb{E}}_{S' \sim D^m} Z_P(S')}{\delta}. \tag{2}
$$

To bound $\mathbb{E}_{S' \sim D^m} Z_P(S')$, let $X_i := \boldsymbol{R}_{\{(x_i, y_i)'\}}(h) \in \triangle_M$ for $i \in [m]$, where $(x_i, y_i)'$ is the $i$'th element of the dummy sample $S'$. Noting that each $X_i$ has mean $\boldsymbol{R}_D(h)$, define the random vectors $X'_i \sim \mathrm{Mult}(1, M, \boldsymbol{R}_D(h))$ and $Y := \sum_{i=1}^{m} X'_i \sim \mathrm{Mult}(m, M, \boldsymbol{R}_D(h))$. Finally let $f : \triangle_M^m \to \mathbb{R}$ be defined by $f(x_1, \ldots, x_m) := \exp\big(\beta d\big(\frac{1}{m} \sum_{i=1}^{m} x_i, \boldsymbol{R}_D(h)\big)\big)$, which is convex since the average is linear, $d$ is convex and the exponential is non-decreasing and convex. Then, by swapping expectations (which is permitted by Fubini's theorem since the argument is non-negative) and applying Lemma 5, we have that $\mathbb{E}_{S' \sim D^m} Z_P(S')$ can be written as

$$
\begin{aligned}
\mathbb{E}_{S' \sim D^m} Z_P(S') &= \mathop{\mathbb{E}}_{S' \sim D^m} \mathop{\mathbb{E}}_{h \sim P} \exp\big(\beta d\big(\boldsymbol{R}_{S'}(h), \boldsymbol{R}_D(h)\big)\big) \\
&= \mathop{\mathbb{E}}_{h \sim P} \mathop{\mathbb{E}}_{S' \sim D^m} \exp\big(\beta d\big(\boldsymbol{R}_{S'}(h), \boldsymbol{R}_D(h)\big)\big) \\
&= \mathop{\mathbb{E}}_{h \sim P} \mathop{\mathbb{E}}_{X_1, \ldots, X_m} \exp\left(\beta d\left(\frac{1}{m} \sum_{i=1}^{m} X_i, \boldsymbol{R}_D(h)\right)\right) \\
&\le \mathop{\mathbb{E}}_{h \sim P} \mathop{\mathbb{E}}_{X'_1, \ldots, X'_m} \exp\left(\beta d\left(\frac{1}{m} \sum_{i=1}^{m} X'_i, \boldsymbol{R}_D(h)\right)\right) \\
&= \mathop{\mathbb{E}}_{h \sim P} \mathop{\mathbb{E}}_{Y} \exp\left(\beta d\left(\frac{1}{m} Y, \boldsymbol{R}_D(h)\right)\right) \\
&= \mathop{\mathbb{E}}_{h \sim P} \sum_{\boldsymbol{k} \in S_{m,M}} \mathrm{Mult}\big(\boldsymbol{k}; m, M, \boldsymbol{R}_D(h)\big) \exp\left(\beta d\big(\tfrac{\boldsymbol{k}}{m}, \boldsymbol{R}_D(h)\big)\right) \\
&\le \sup_{\boldsymbol{r} \in \triangle_M} \left[\sum_{\boldsymbol{k} \in S_{m,M}} \mathrm{Mult}\big(\boldsymbol{k}; m, M, \boldsymbol{r}\big) \exp\left(\beta d\big(\tfrac{\boldsymbol{k}}{m}, \boldsymbol{r}\big)\right)\right].
\end{aligned}
$$

Which is the definition of $\mathcal{I}_d(m, \beta)$. Inequality (1) then follows by substituting this bound on $\mathbb{E}_{S' \sim D^m} Z_P(S')$ into (2) and dividing by $\beta$. □

## 3.2 Statement and proof of the generalised corollary

We now apply our generalised theorem with $\beta = m$ and $d(\boldsymbol{q}, \boldsymbol{p}) = \mathrm{kl}(\boldsymbol{q}\|\boldsymbol{p})$. This results in the following corollary, analogous to Corollary 2 (although the multi-dimensionality makes the proof much more involved, requiring multiple lemmas and extra arguments to make the main idea go through). We give two forms of the bound since, while the second is looser, the first is not practical to calculate except when $m$ is very small.

**Corollary 7.** *Let $\mathcal{X}$ and $\mathcal{Y}$ be arbitrary sets and $\bigcup_{j=1}^{M} E_j$ be a disjoint partition of $\mathcal{Y}^2$. Let $D \in \mathcal{M}(\mathcal{X} \times \mathcal{Y})$ be a data-generating distribution and $\mathcal{H} \subseteq \mathcal{M}(\mathcal{Y})^{\mathcal{X}}$ be a hypothesis class. For any prior $P \in \mathcal{M}(\mathcal{H})$, $\delta \in (0, 1]$ and sample size $m$, with probability at least $1 - \delta$ over the random draw $S \sim D^m$, we have that simultaneously for all posteriors $Q \in \mathcal{M}(\mathcal{H})$*

$$\mathrm{kl}\big(\boldsymbol{R}_S(Q)\|\boldsymbol{R}_D(Q)\big) \leq \frac{1}{m}\left[\mathrm{KL}(Q\|P) + \ln\left(\frac{m!}{\delta m^m}\sum_{\boldsymbol{k} \in S_{m,M}}\prod_{j=1}^{M}\frac{k_j^{k_j}}{k_j!}\right)\right] \tag{3}$$

$$\leq \frac{1}{m}\left[\mathrm{KL}(Q\|P) + \ln\left(\frac{1}{\delta}\sqrt{\pi}e^{1/(12m)}\left(\frac{m}{2}\right)^{\frac{M-1}{2}}\sum_{z=0}^{M-1}\binom{M}{z}\frac{1}{(\pi m)^{z/2}\,\Gamma\left(\frac{M-z}{2}\right)}\right)\right], \tag{4}$$

*where the second inequality holds provided $m \geq M$. Further, the bounds are unchanged if one restricts to an ordinary hypothesis class, namely if $\mathcal{H} \subseteq \mathcal{Y}^{\mathcal{X}}$.*

While analogous corollaries can be obtained from Theorem 3 by other choices of convex function $d$, the kl-divergence leads to convenient cancellations that remove the dependence of $\mathcal{I}_{\mathrm{kl}}(m, \beta, \boldsymbol{r})$ on $\boldsymbol{r}$, making $\mathcal{I}_{\mathrm{kl}}(m, \beta) := \sup_{\boldsymbol{r} \in \triangle_M}\mathcal{I}_{\mathrm{kl}}(m, \beta, \boldsymbol{r})$ simple to evaluate. Note (4) is logarithmic in $1/\delta$ (typical of PAC-Bayes bounds) and thus the confidence can be increased very cheaply. Ignoring logarithmic terms, (4) is $\mathcal{O}(1/m)$, also as expected. As for $M$, a simple analysis shows that (4) grows only sublinearly in $M$, meaning $M$ can be made quite large provided one has a reasonable amount of data. To prove Corollary 7 we require Lemma 8, the proof of which is deferred to Appendix B.2.

**Lemma 8.** *For integers $M \geq 1$ and $m \geq M$, $\sum_{\boldsymbol{k} \in S_{m,M}^{>0}}\frac{1}{\prod_{j=1}^{M}\sqrt{k_j}} \leq \frac{\pi^{\frac{M}{2}}m^{\frac{M-2}{2}}}{\Gamma(\frac{M}{2})}$.*

*Proof.* (of Corollary 7) Applying Theorem 3 with $d(\boldsymbol{q}, \boldsymbol{p}) = \mathrm{kl}(\boldsymbol{q}\|\boldsymbol{p})$ (defined in Section 2) and $\beta = m$ gives that with probability at least $1 - \delta$ over $S \sim D^m$, simultaneously for all posteriors $Q \in \mathcal{M}(\mathcal{H})$, $\mathrm{kl}\big(\boldsymbol{R}_S(Q)\|\boldsymbol{R}_D(Q)\big) \leq \frac{1}{m}[\mathrm{KL}(Q\|P) + \ln\frac{\mathcal{I}_{\mathrm{kl}}(m,m)}{\delta}]$, where $\mathcal{I}_{\mathrm{kl}}(m, m) := \sup_{\boldsymbol{r} \in \triangle_M}[\sum_{\boldsymbol{k} \in S_{m,M}}\mathrm{Mult}(\boldsymbol{k}; m, M, \boldsymbol{r})\exp\big(m\mathrm{kl}(\frac{\boldsymbol{k}}{m}, \boldsymbol{r})\big)]$. Thus, to establish the first inequality of the corollary, it suffices to show that

$$\mathcal{I}_{\mathrm{kl}}(m, m) \leq \frac{m!}{m^m}\sum_{\boldsymbol{k} \in S_{m,M}}\prod_{j=1}^{M}\frac{k_j^{k_j}}{k_j!}. \tag{5}$$

To see this, for each fixed $\boldsymbol{r} = (r_1, \ldots, r_M) \in \triangle_M$ let $J_{\boldsymbol{r}} = \{j \in [M] : r_j = 0\}$. Then $\mathrm{Mult}(\boldsymbol{k}; m, M, \boldsymbol{r}) = 0$ for any $\boldsymbol{k} \in S_{m,M}$ such that $k_j \neq 0$ for some $j \in J_{\boldsymbol{r}}$. For the other $\boldsymbol{k} \in S_{m,M}$, namely those such that $k_j = 0$ for all $j \in J_{\boldsymbol{r}}$, the probability term can be written as $\mathrm{Mult}(\boldsymbol{k}; m, M, \boldsymbol{r}) = \frac{m!}{\prod_{j=1}^{M}k_j!}\prod_{j=1}^{M}r_j^{k_j} = \frac{m!}{\prod_{j \notin J_{\boldsymbol{r}}}k_j!}\prod_{j \notin J_{\boldsymbol{r}}}r_j^{k_j}$, and (recalling the convention that $0\ln\frac{0}{0} = 0$) the term $\exp(m\mathrm{kl}(\frac{\boldsymbol{k}}{m}, \boldsymbol{r}))$ can be written as

$$\exp\left(m\sum_{j=1}^{M}\frac{k_j}{m}\ln\frac{\frac{k_j}{m}}{r_j}\right) = \exp\left(\sum_{j \notin J_{\boldsymbol{r}}}k_j\ln\frac{k_j}{mr_j}\right) = \prod_{j \notin J_{\boldsymbol{r}}}\left(\frac{k_j}{mr_j}\right)^{k_j} = \frac{1}{m^m}\prod_{j \notin J_{\boldsymbol{r}}}\left(\frac{k_j}{r_j}\right)^{k_j},$$

where the last equality is obtained by recalling that the $k_j$ sum to $m$. Substituting these two expressions into the definition of $\mathcal{I}_{\mathrm{kl}}(m, m)$ and only summing over those $\boldsymbol{k} \in S_{m,M}$ with non-zero probability, we obtain

$$\sum_{\boldsymbol{k} \in S_{m,M}}\mathrm{Mult}(\boldsymbol{k}; m, M, \boldsymbol{r})\exp\big(m\mathrm{kl}\left(\tfrac{\boldsymbol{k}}{m}, \boldsymbol{r}\right)\big) = \sum_{\substack{\boldsymbol{k} \in S_{m,M}: \\ \forall j \in J_{\boldsymbol{r}}\ k_j = 0}}\mathrm{Mult}(\boldsymbol{k}; m, M, \boldsymbol{r})\exp\big(m\mathrm{kl}\left(\tfrac{\boldsymbol{k}}{m}, \boldsymbol{r}\right)\big)$$

$$
= \sum_{\substack{\boldsymbol{k}\in S_{m,M}: \\ \forall j\in J_{\boldsymbol{r}} \; k_j=0}} \frac{m!}{\prod_{j\notin J_{\boldsymbol{r}}} k_j!} \prod_{j\notin J_{\boldsymbol{r}}} r_j^{k_j} \frac{1}{m^m} \prod_{j\notin J_{\boldsymbol{r}}} \left(\frac{k_j}{r_j}\right)^{k_j}
$$

$$
= \frac{m!}{m^m} \sum_{\substack{\boldsymbol{k}\in S_{m,M}: \\ \forall j\in J_{\boldsymbol{r}} \; k_j=0}} \prod_{j\notin J_{\boldsymbol{r}}} \frac{k_j^{k_j}}{k_j!}
$$

$$
= \frac{m!}{m^m} \sum_{\substack{\boldsymbol{k}\in S_{m,M}: \\ \forall j\in J_{\boldsymbol{r}} \; k_j=0}} \prod_{j=1}^{M} \frac{k_j^{k_j}}{k_j!} \qquad (\text{because } \tfrac{0^0}{0!}=1)
$$

$$
\leq \frac{m!}{m^m} \sum_{\boldsymbol{k}\in S_{m,M}} \prod_{j=1}^{M} \frac{k_j^{k_j}}{k_j!}.
$$

Since this is independent of $\boldsymbol{r}$, it also holds after taking the supremum over $\boldsymbol{r}\in \triangle_M$ of the left hand side. We have thus established (5) and hence (3). Now, defining $f:\bigcup_{M=2}^{\infty} S_{m,M} \to \mathbb{R}$ by $f(\boldsymbol{k}) = \prod_{j=1}^{|\boldsymbol{k}|} k_j^{k_j}/k_j!$, we see that to establish inequality (4) it suffices to show that

$$
\frac{m!}{m^m} \sum_{\boldsymbol{k}\in S_{m,M}} f(\boldsymbol{k}) \leq \sqrt{\pi} e^{1/12m} \left(\frac{m}{2}\right)^{\frac{M-1}{2}} \sum_{z=0}^{M-1} \binom{M}{z} \frac{1}{(\pi m)^{z/2} \, \Gamma\left(\frac{M-z}{2}\right)}. \tag{6}
$$

We show this by upper bounding each $f(\boldsymbol{k})$ individually using Stirling's formula: $\forall n\geq 1$ $\sqrt{2\pi n}\left(\frac{n}{e}\right)^n < n! < \sqrt{2\pi n}\left(\frac{n}{e}\right)^n e^{\frac{1}{12n}}$. Since we cannot use this to upper bound $1/k_j!$ when $k_j=0$, we partition the sum above according to the number of coordinates of $\boldsymbol{k}$ at which $k_j=0$. Let $z$ index the number of such coordinates. Since $f$ is symmetric under permutations of its arguments,

$$
\sum_{\boldsymbol{k}\in S_{m,M}} f(\boldsymbol{k}) = \sum_{z=0}^{M-1} \binom{M}{z} \sum_{\boldsymbol{k}\in S_{m,M-z}^{>0}} f(\boldsymbol{k}). \tag{7}
$$

For $\boldsymbol{k}\in S_{m,M}^{>0}$ Stirling's formula yields $f(\boldsymbol{k}) \leq \prod_{j=1}^{M} \frac{k_j^{k_j}}{\sqrt{2\pi k_j}\left(\frac{k_j}{e}\right)^{k_j}} = \prod_{j=1}^{M} \frac{e^{k_j}}{\sqrt{2\pi k_j}} = \frac{e^m}{(2\pi)^{M/2}} \prod_{j=1}^{M} \frac{1}{\sqrt{k_j}}$. An application of Lemma 8 now gives

$$
\sum_{\boldsymbol{k}\in S_{m,M-z}^{>0}} f(\boldsymbol{k}) \leq \frac{e^m}{(2\pi)^{M/2}} \sum_{\boldsymbol{k}\in S_{m,M-z}^{>0}} \prod_{j=1}^{M} \frac{1}{\sqrt{k_j}} \leq \frac{e^m}{(2\pi)^{\frac{M}{2}}} \frac{\pi^{\frac{M-z}{2}} m^{\frac{M-z-2}{2}}}{\Gamma\left(\frac{M-z}{2}\right)} = \frac{e^m m^{\frac{M-2}{2}}}{2^{\frac{M}{2}} (\pi m)^{z/2} \, \Gamma\left(\frac{M-z}{2}\right)}.
$$

Substituting this into equation (7) and bounding $m!$ using Stirling's formula, we have

$$
\frac{m!}{m^m} \sum_{\boldsymbol{k}\in S_{m,M}} f(\boldsymbol{k}) \leq \frac{\sqrt{2\pi m} e^{1/12m}}{e^m} \sum_{z=0}^{M-1} \binom{M}{z} \frac{e^m m^{\frac{M-2}{2}}}{2^{M/2} (\pi m)^{z/2} \, \Gamma\left(\frac{M-z}{2}\right)}
$$

$$
= \sqrt{\pi} e^{1/12m} \left(\frac{m}{2}\right)^{\frac{M-1}{2}} \sum_{z=0}^{M-1} \binom{M}{z} \frac{1}{(\pi m)^{z/2} \, \Gamma\left(\frac{M-z}{2}\right)}
$$

which is (6), establishing (4) and therefore completing the proof. $\qquad \square$

## 4 Implied Bounds and Construction of a Differentiable Training Objective

As already discussed, a multitude of bounds can be derived from Theorem 3 and Corollary 7, all of which then hold simultaneously with high probability. For example, suppose after a use of Corollary 7 we have a bound of the form $\mathrm{kl}(\boldsymbol{R}_S(Q)\|\boldsymbol{R}_D(Q)) \leq B$. The following proposition then yields the bounds $L_j \leq R_D^j(Q) \leq U_j$, where $L_j := \inf\{p\in[0,1]: \mathrm{kl}(R_S^j(Q)\|p) \leq B\}$ and $U_j := \sup\{p\in [0,1]: \mathrm{kl}(R_S^j(Q)\|p) \leq B\}$. Moreover, since in the worst case we have $\mathrm{kl}(\boldsymbol{R}_S(Q)\|\boldsymbol{R}_D(Q)) = B$,

the proposition shows that the lower and upper bounds $L_j$ and $U_j$ are the tightest possible, since if $R_D^j(Q) \notin [L_j, U_j]$ then $\mathrm{kl}(R_S^j(Q)\|R_D^j(Q)) > B$ implying $\mathrm{kl}(\boldsymbol{R}_S(Q)\|\boldsymbol{R}_D(Q)) > B$. For a more precise version of this argument and a proof of Proposition 9, see Appendix B.3.

**Proposition 9.** *Let $\boldsymbol{q}, \boldsymbol{p} \in \triangle_M$. Then $\mathrm{kl}(q_j\|p_j) \leq \mathrm{kl}(\boldsymbol{q}\|\boldsymbol{p})$ for all $j \in [M]$, with equality when $p_i = \frac{1-p_j}{1-q_j} q_i$. for all $i \neq j$.*

As a second much more interesting example, suppose we can quantify how bad an error of each type is by means of a loss vector $\boldsymbol{\ell} \in [0, \infty)^M$, where $\ell_j$ is the loss we attribute to an error of type $E_j$. We may then be interested in bounding the *total risk* $R_D^T(Q) \in [0, \infty)$ of $Q$ which, recall, is defined by $R_D^T(Q) := \sum_{j=1}^M \ell_j R_D^j(Q)$. Indeed, given a bound of the form $\mathrm{kl}(\boldsymbol{R}_S(Q)\|\boldsymbol{R}_D(Q)) \leq B$, we can derive $R_D^T(Q) \leq \sup\{\sum_{j=1}^M \ell_j r_j : \boldsymbol{r} \in \triangle_M, \mathrm{kl}(\boldsymbol{R}_S(Q)\|\boldsymbol{r}) \leq B\}$. This motivates the following definition of $\mathrm{kl}_{\boldsymbol{\ell}}^{-1}(\boldsymbol{u}|c)$. To see that this is indeed well-defined (at least when $\boldsymbol{u} \in \triangle_M^{>0}$), see the discussion at the beginning of Appendix B.4.

**Definition 10.** *For $\boldsymbol{u} \in \triangle_M, c \in [0, \infty)$ and $\boldsymbol{\ell} \in [0, \infty)^M$, define $\mathrm{kl}_{\boldsymbol{\ell}}^{-1}(\boldsymbol{u}|c) = \sup\{\sum_{j=1}^M \ell_j v_j : \boldsymbol{v} \in \triangle_M, \mathrm{kl}(\boldsymbol{u}\|\boldsymbol{v}) \leq c\}$.*

Can we calculate $\mathrm{kl}_{\boldsymbol{\ell}}^{-1}(\boldsymbol{u}|c)$ and hence $f_{\boldsymbol{\ell}}(\mathrm{kl}_{\boldsymbol{\ell}}^{-1}(\boldsymbol{u}|c))$ in order to evaluate the bound on the total risk? Additionally, if we wish to use the bound on the total risk as a training objective, can we calculate the partial derivatives of $f_{\boldsymbol{\ell}}^*(\boldsymbol{u}, c) := f_{\boldsymbol{\ell}}(\mathrm{kl}_{\boldsymbol{\ell}}^{-1}(\boldsymbol{u}|c))$ with respect to the $u_j$ and $c$ so that we can use gradient descent? Our Proposition 11 answers both of these questions in the affirmative, at least in the sense that it provides a speedy method for approximating these quantities to arbitrary precision provided $u_j > 0$ for all $j \in [M]$ and $c > 0$. Indeed, the only approximation step required is that of approximating the unique root of a continuous and strictly increasing scalar function. Thus, provided the $u_j$ themselves are differentiable, Corollary 7 combined with Proposition 11 yields a tractable and fully differentiable objective that can be used for training. More details on how this can be done, including an algorithm written in pseudocode, can be found in Appendix A. While somewhat analogous to the technique used in [9] to obtain derivatives of the one-dimensional kl-inverse, our proposition directly yields derivatives on the total risk by (implicitly) employing the envelope theorem (see for example 34). Since the proof of Proposition 11 is rather long and technical, we defer it to Appendix B.4.

**Proposition 11.** *Fix $\boldsymbol{\ell} \in [0, \infty)^M$ such that not all $\ell_j$ are equal, and define $f_{\boldsymbol{\ell}} : \triangle_M \to [0, \infty)$ by $f_{\boldsymbol{\ell}}(\boldsymbol{v}) := \sum_{j=1}^M \ell_j v_j$. For all $\tilde{\boldsymbol{u}} = (\boldsymbol{u}, c) \in \triangle_M^{>0} \times (0, \infty)$, define $\boldsymbol{v}^*(\tilde{\boldsymbol{u}}) := \mathrm{kl}_{\boldsymbol{\ell}}^{-1}(\boldsymbol{u}|c) \in \triangle_M$ and let $\mu^*(\tilde{\boldsymbol{u}}) \in (-\infty, -\max_j \ell_j)$ be the unique solution to $c = \phi_{\boldsymbol{\ell}}(\mu)$, where $\phi_{\boldsymbol{\ell}} : (-\infty, -\max_j \ell_j) \to \mathbb{R}$ is given by $\phi_{\boldsymbol{\ell}}(\mu) := \ln(-\sum_{j=1}^M \frac{u_j}{\mu+\ell_j}) + \sum_{j=1}^M u_j \ln(-(\mu+\ell_j))$, which is continuous and strictly increasing. Then $\boldsymbol{v}^*(\tilde{\boldsymbol{u}}) = \mathrm{kl}_{\boldsymbol{\ell}}^{-1}(\boldsymbol{u}|c)$ is given by*

$$\boldsymbol{v}^*(\tilde{\boldsymbol{u}})_j = \frac{\lambda^*(\tilde{\boldsymbol{u}})u_j}{\mu^*(\tilde{\boldsymbol{u}})+\ell_j} \quad \textit{for } j \in [M], \quad \textit{where} \quad \lambda^*(\tilde{\boldsymbol{u}}) = \left(\sum_{j=1}^M \frac{u_j}{\mu^*(\tilde{\boldsymbol{u}})+\ell_j}\right)^{-1}.$$

*Further, defining $f_{\boldsymbol{\ell}}^* : \triangle_M^{>0} \times (0, \infty) \to [0, \infty)$ by $f_{\boldsymbol{\ell}}^*(\tilde{\boldsymbol{u}}) := f_{\boldsymbol{\ell}}(\boldsymbol{v}^*(\tilde{\boldsymbol{u}}))$, we have that*

$$\frac{\partial f_{\boldsymbol{\ell}}^*}{\partial u_j}(\tilde{\boldsymbol{u}}) = \lambda^*(\tilde{\boldsymbol{u}})\left(1 + \ln \frac{u_j}{\boldsymbol{v}^*(\tilde{\boldsymbol{u}})_j}\right) \qquad \textit{and} \qquad \frac{\partial f_{\boldsymbol{\ell}}^*}{\partial c}(\tilde{\boldsymbol{u}}) = -\lambda^*(\tilde{\boldsymbol{u}}).$$

## 5 Perspectives

By abstracting to a general setting of discretised error types, we established a novel type of generalisation bound (Theorem 3) providing far richer information than existing PAC-Bayes bounds. Through our Corollary 7 and Proposition 11, our bound inspires a training algorithm (see Appendix A) suitable for many different learning problems, including structured output prediction [as investigated by 8, in the PAC-Bayes setting], multi-task learning and learning-to-learn [see *e.g.* 23]. We will demonstrate these applications and our bound's utility for real-world learning problems in an empirical follow-up study. Note we require i.i.d. data, which in practice is frequently not the case or is hard to verify. Further, the number of error types $M$ must be finite. While in continuous scenarios it would be preferable to be able to quantify the entire distribution of loss values without having to discretise into finitely many error types, in the multiclass setting our framework is entirely suitable.

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
