## A Recipe for implementing and deploying our strategy

We here outline more explicitly how Corollary 7 and Proposition 11 may be used to formulate a fully differentiable objective by which a model may be trained.

First, if one wishes to make hard labels, namely $\mathcal{H} \subseteq \mathcal{Y}^{\mathcal{X}}$, it will first be necessary to use a surrogate class of soft hypotheses $\mathcal{H}' \subseteq \mathcal{M}(\mathcal{Y})^{\mathcal{X}}$ during training, before reverting to hard labels for example by taking the mean label or the one with highest probability. Using soft hypotheses during training is necessary to ensure that the empirical $j$-risks $R_S^j(Q)$ are differentiable in the model parameters. Since how one chooses to do this will depend on the specific use case, we restrict our attention here to the case of soft hypotheses. Specifically, we consider a class of soft hypotheses $\mathcal{H} = \{h_\theta : \theta \in \mathbb{R}^N\} \subseteq \mathcal{M}(\mathcal{Y})^{\mathcal{X}}$ parameterised by the weights $\theta \in \mathbb{R}^N$ of some neural network of a given architecture with $N$ parameters in such a way that the $R_S^j(h_\theta)$ are differentiable in $\theta$. A concrete example would be multiclass classification using a fully connected neural network with output being softmax probabilities on the classes so that the $R_S^j(h_\theta)$ are differentiable.

Second, it is necessary to restrict the prior and posterior $P, Q \in \mathcal{M}(\mathcal{H})$ to a parameterised subset of $\mathcal{M}(\mathcal{H})$ in which $\mathrm{KL}(Q\|P)$ has a closed form which is differentiable in the parameterisation. A simple choice for our case of a neural network with $N$ parameters is $P, Q \in \{\mathcal{N}(\boldsymbol{w}, \mathrm{diag}(\boldsymbol{s})) : \boldsymbol{w} \in \mathbb{R}^N, \boldsymbol{s} \in \mathbb{R}^N_{>0}\}$. For prior a $P_{\boldsymbol{v},\boldsymbol{r}} = \mathcal{N}(\boldsymbol{v}, \mathrm{diag}(\boldsymbol{r}))$ and posterior $Q_{\boldsymbol{w},\boldsymbol{s}} = \mathcal{N}(\boldsymbol{w}, \mathrm{diag}(\boldsymbol{s}))$ we have the closed form

$$\mathrm{KL}(Q_{\boldsymbol{w},\boldsymbol{s}}\|P_{\boldsymbol{v},\boldsymbol{r}}) = \frac{1}{2}\left[\sum_{n=1}^N \left(\frac{s_n}{r_n} + \frac{(w_n - v_n)^2}{r_n} + \ln\frac{r_n}{s_n}\right) - N\right],$$

which is indeed differentiable in $\boldsymbol{v}, \boldsymbol{r}, \boldsymbol{w}$ and $\boldsymbol{s}$. While $Q_{\boldsymbol{w},\boldsymbol{s}}$ and $P_{\boldsymbol{v},\boldsymbol{r}}$ are technically distributions on $\mathbb{R}^D$ rather than $\mathcal{H}$, the KL-divergence between the distributions they induce on $\mathcal{H}$ will be at most as large as the expression above. Thus, substituting the expression above into the bounds we prove in Section 3 can only increase the value of the bounds, meaning the enlarged bounds certainly still hold with probability at least $1 - \delta$.

Third, in all but the simplest cases $R_S^j(Q_{\boldsymbol{w},\boldsymbol{s}})$ will not have a closed form, much less one that is differentiable in $\boldsymbol{w}$ and $\boldsymbol{s}$. A common solution to this is to use the so-called pathwise gradient estimator. In our case, this corresponds to drawing $\boldsymbol{\epsilon} \sim \mathcal{N}(\boldsymbol{0}, \mathbb{I})$, where $\mathbb{I}$ is the $N \times N$ identity matrix, and estimating

$$\nabla_{\boldsymbol{w},\boldsymbol{s}} R_S^j(Q_{\boldsymbol{w},\boldsymbol{s}}) = \nabla_{\boldsymbol{w},\boldsymbol{s}}\left[\mathbb{E}_{\boldsymbol{\epsilon}'\sim\mathcal{N}(\boldsymbol{0},\mathbb{I})} R_S^j(h_{\boldsymbol{w}+\boldsymbol{\epsilon}'\odot\sqrt{\boldsymbol{s}}})\right] \approx \nabla_{\boldsymbol{w},\boldsymbol{s}} R_S^j(h_{\boldsymbol{w}+\boldsymbol{\epsilon}\odot\sqrt{\boldsymbol{s}}}),$$

where $h_{\boldsymbol{w}}$ denotes the function expressed by the neural network with parameters $\boldsymbol{w}$. For a proof that this is an unbiased estimator, and for other methods for estimating the gradients of expectations, see the survey [26].

Fourth, one must choose the prior. Designing priors which are optimal in some sense (*i.e.*, minimising the Kullback-Leibler term in the right-hand side of generalisation bounds) has been at the core of an active line of work in the PAC-Bayesian literature. For the sake of simplicity, and since it is out of the scope of our contributions, we assume here that the prior is given beforehand, although we stress that practitioners should pay great attention to its tuning. For our purposes, it suffices to say that if one is using a data-dependent prior then it is necessary to partition the sample into $S = S_{\mathrm{Prior}} \cup S_{\mathrm{Bound}}$, where $S_{\mathrm{Prior}}$ is used to train the prior and $S_{\mathrm{Bound}}$ is used to evaluate the bound. Since our bound holds uniformly over posteriors $Q \in \mathcal{M}(\mathcal{H})$, the entire sample $S$ is free to be used to train the posterior $Q$. For a more in-depth discussion on the choice of prior, we refer to the following body of work: Ambroladze et al. [2], Lever et al. [20, 21], Parrado-Hernández et al. [29], Dziugaite and Roy [13, 14], Rivasplata et al. [32], Letarte et al. [19], Pérez-Ortiz et al. [30], Dziugaite et al. [12], Biggs and Guedj [4, 6, 5].

Finally, given a confidence level $\delta \in (0, 1]$, one may use Algorithm 1 to obtain a posterior $Q_{\boldsymbol{w},\boldsymbol{s}}$ with minimal upper bound on the total risk. Note we take the pointwise logarithm of the variances $\boldsymbol{r}$ and $\boldsymbol{s}$ to obtain unbounded parameters on which to perform stochastic gradient descent or some other minimisation algorithm. We use $\oplus$ to denote vector concatenation. The algorithm can be straightforwardly adapted to permit mini-batches by, for each epoch, sequentially repeating the steps with $S$ equal to each mini-batch.

**Input:**
```
𝒳, 𝒴 /* Arbitrary input and output spaces                                    */
⋃_{j=1}^{M} E_j = 𝒴² /* A finite partition into error types                  */
ℓ ∈ [0,∞)^M /* A vector of losses, not all equal                            */
S = S_Prior ∪ S_Bound ∈ (𝒳 × 𝒴)^m /* A partitioned i.i.d.  sample          */
N ∈ ℕ /* The number of model parameters                                      */
P_{v,r}, v(S_Prior) ∈ ℝ^N, r(S_Prior) ∈ ℝ_{≥0}^N /* A (data-dependent) prior */
Q_{w_0,s_0}, w_0 ∈ ℝ^N, s_0 ∈ ℝ_{≥0}^N /* An initial posterior             */
δ ∈ (0,1] /* A confidence level                                              */
λ > 0 /* A learning rate                                                     */
T /* The number of epochs to train for                                       */
```

**Output:**
```
Q_{w,s}, w ∈ ℝ^N, s ∈ ℝ_{≥0}^N /* A trained posterior                      */
```

**Procedure:**
$\boldsymbol{\zeta}_0 \leftarrow \log \boldsymbol{s}_0$ /* Transform to unbounded scale parameters ... */
$\boldsymbol{p} \leftarrow \boldsymbol{w}_0 \oplus \boldsymbol{\zeta}_0$ /* Collect mean and scale parameters ... */
**for** $t \leftarrow 1$ **to** $T$ **do**

$\quad$ Draw $\boldsymbol{\epsilon} \sim \mathcal{N}(\mathbf{0}, \mathbb{I})$

$\quad \boldsymbol{u} \leftarrow \boldsymbol{R}_S \left( h_{\boldsymbol{w}+\boldsymbol{\epsilon}\odot\sqrt{\exp(\boldsymbol{\zeta})}} \right)$

$\quad B \leftarrow$
$\quad \frac{1}{m} \left[ \mathrm{KL}\left( Q_{\boldsymbol{w},\exp(\boldsymbol{\zeta})} \big\| P_{\boldsymbol{v},\boldsymbol{r}} \right) + \ln \left( \frac{1}{\delta} \sqrt{\pi} e^{1/12m} \left( \frac{m}{2} \right)^{\frac{M-1}{2}} \sum_{z=0}^{M-1} \binom{M}{z} \frac{1}{(\pi m)^{z/2}\Gamma\left( \frac{M-z}{2} \right)} \right) \right]$

$\quad \tilde{\boldsymbol{u}} \leftarrow (u_1, \ldots, u_M, B)$
$\quad \boldsymbol{G} \leftarrow \mathbf{0}_{2N\times(M+1)}$ /* Initialise gradient matrix ... */
$\quad \boldsymbol{F} \leftarrow \mathbf{0}_{M+1}$ /* Initialise gradient vector ... */
$\quad$ **for** $j \leftarrow 1$ **to** $M+1$ **do**

$\quad\quad \boldsymbol{F}_j \leftarrow \frac{\partial f_{\boldsymbol{\ell}}^*}{\partial \tilde{u}_j}(\tilde{\boldsymbol{u}})$ /* Gradients of total loss from Prop 11 ... */
$\quad\quad$ **for** $i \leftarrow 1$ **to** $2N$ **do**

$\quad\quad\quad \boldsymbol{G}_{i,j} \leftarrow \frac{\partial \tilde{u}_j}{\partial p_i}(\boldsymbol{p})$ /* Gradients of empirical risks and bound ... */
$\quad\quad$ **end**

$\quad$ **end**
$\quad \boldsymbol{H} \leftarrow \boldsymbol{G}\boldsymbol{F}$ /* Gradients of total loss w.r.t.  parameters ... */
$\quad \boldsymbol{p} \leftarrow \boldsymbol{p} - \lambda\boldsymbol{H}$ /* Gradient step ... */

**end**
$\boldsymbol{w} = (p_1, \ldots, p_N)$
$\boldsymbol{s} = (\exp(p_{N+1}), \ldots, \exp(p_{2N}))$
**return** $\boldsymbol{w}, \boldsymbol{s}$

**Algorithm 1:** Calculating a posterior with minimal bound on the total risk.

## B  Proofs

### B.1  Proof of Lemma 5

Let $\boldsymbol{E}_M := \{\boldsymbol{e}_1, \ldots, \boldsymbol{e}_M\}$, namely the set of $M$-dimensional basis vectors. We will denote a typical element of $\boldsymbol{E}_M^m$ by $\boldsymbol{\eta}^{(m)} = (\boldsymbol{\eta}_1, \ldots, \boldsymbol{\eta}_m)$. For any $\boldsymbol{x}^{(m)} = (\boldsymbol{x}_1, \ldots, \boldsymbol{x}_m) \in \triangle_M^m$, a straightforward induction on $m$ yields

$$\sum_{\boldsymbol{\eta}^{(m)} \in \boldsymbol{E}_M^m} \left( \prod_{i=1}^{m} \boldsymbol{x}_i \cdot \boldsymbol{\eta}_i \right) = 1. \tag{8}$$

To see this, for $m = 1$ we have $\boldsymbol{E}_M^1 = \{(\boldsymbol{e}_1,), \ldots, (\boldsymbol{e}_M,)\}$, where we have been pedantic in using 1-tuples to maintain consistency with larger values of $m$. Thus, for any $\boldsymbol{x}^{(1)} = (\boldsymbol{x}_1,) \in \triangle_M^1$, the left

hand side of equation (8) can be written as

$$\sum_{j=1}^{M} \boldsymbol{x}_1 \cdot \boldsymbol{e}_j = \sum_{j=1}^{M} (\boldsymbol{x}_1)_j = 1.$$

Now suppose that equation (8) holds for any $\boldsymbol{x}^{(m)} \in \triangle_M^m$ and let $\boldsymbol{x}^{(m+1)} = (\boldsymbol{x}_1, \ldots, \boldsymbol{x}_{m+1}) \in \triangle_M^{m+1}$. Then the left hand side of equation (8) can be written as

$$\sum_{\boldsymbol{\eta}^{(m+1)} \in \boldsymbol{E}_M^{m+1}} \left( \prod_{i=1}^{m+1} \boldsymbol{x}_i \cdot \boldsymbol{\eta}_i \right) = \sum_{\boldsymbol{\eta}^{(m)} \in \boldsymbol{E}_M^m} \sum_{j=1}^{M} \left( \prod_{i=1}^{m} \boldsymbol{x}_i \cdot \boldsymbol{\eta}_i \right) (\boldsymbol{x}_{m+1} \cdot \boldsymbol{e}_j)$$

$$= \sum_{\boldsymbol{\eta}^{(m)} \in \boldsymbol{E}_M^m} \left( \prod_{i=1}^{m} \boldsymbol{x}_i \cdot \boldsymbol{\eta}_i \right) \sum_{j=1}^{M} (\boldsymbol{x}_{m+1} \cdot \boldsymbol{e}_j) = 1.$$

We now show that any $\boldsymbol{x}^{(m)} = (\boldsymbol{x}_1, \ldots, \boldsymbol{x}_m) \in \triangle_M^m$ can be written as a convex combination of the elements of $\boldsymbol{E}_M^m$ in the following way

$$\boldsymbol{x}^{(m)} = \sum_{\boldsymbol{\eta}^{(m)} \in \boldsymbol{E}_M^m} \left( \prod_{i=1}^{m} \boldsymbol{x}_i \cdot \boldsymbol{\eta}_i \right) \boldsymbol{\eta}^{(m)}. \tag{9}$$

We have already shown that the weights sum to one, and they are clearly elements of $[0, 1]$, so the right hand side of equation (9) is indeed a convex combination of the elements of $\boldsymbol{E}_M^m$. We now show that equation (9) holds, again by induction.

For $m = 1$ and any $\boldsymbol{x}^{(1)} = (\boldsymbol{x}_1, ) \in \triangle_M^1$, the right hand side of equation (9) can be written as

$$\sum_{j=1}^{M} (\boldsymbol{x}_1 \cdot \boldsymbol{e}_j)(\boldsymbol{e}_j, ) = (\boldsymbol{x}_1, ) = \boldsymbol{x}.$$

For the inductive hypothesis, suppose equation (9) holds for some arbitrary $m \geq 1$, and denote elements of $\boldsymbol{E}_M^{m+1}$ by $\boldsymbol{\eta}^{(m)} \oplus (\boldsymbol{e}, )$ for some $\boldsymbol{\eta}^{(m)} \in \boldsymbol{E}_M^m$ and $\boldsymbol{e} \in \boldsymbol{E}_M$, where $\oplus$ denotes vector concatenation. Then for any $\boldsymbol{x}^{(m+1)} = \boldsymbol{x}^{(m)} \oplus (\boldsymbol{x}_{m+1}, ) = (\boldsymbol{x}_1, \ldots, \boldsymbol{x}_{m+1}) \in \triangle_M^{m+1}$, the right hand side of equation (9) can be written as

$$\sum_{\boldsymbol{\eta}^{(m+1)} \in \boldsymbol{E}_M^{m+1}} \left( \prod_{i=1}^{m+1} \boldsymbol{x}_i \cdot \boldsymbol{\eta}_i \right) \boldsymbol{\eta}^{(m+1)} = \sum_{\boldsymbol{\eta}^{(m)} \in \boldsymbol{E}_M^m} \sum_{j=1}^{M} \left( \prod_{i=1}^{m} \boldsymbol{x}_i \cdot \boldsymbol{\eta}_i \right) (\boldsymbol{x}_{m+1} \cdot \boldsymbol{e}_j) \boldsymbol{\eta}^{(m)} \oplus (\boldsymbol{e}_j, )$$

$$= \sum_{\boldsymbol{\eta}^{(m)} \in \boldsymbol{E}_M^m} \sum_{j=1}^{M} \left( \prod_{i=1}^{m} \boldsymbol{x}_i \cdot \boldsymbol{\eta}_i \right) (\boldsymbol{x}_{m+1} \cdot \boldsymbol{e}_j) \boldsymbol{\eta}^{(m)}$$

$$\oplus \sum_{\boldsymbol{\eta}^{(m)} \in \boldsymbol{E}_M^m} \sum_{j=1}^{M} \left( \prod_{i=1}^{m} \boldsymbol{x}_i \cdot \boldsymbol{\eta}_i \right) (\boldsymbol{x}_{m+1} \cdot \boldsymbol{e}_j)(\boldsymbol{e}_j, )$$

$$= \sum_{j=1}^{M} (\boldsymbol{x}_{m+1} \cdot \boldsymbol{e}_j) \sum_{\boldsymbol{\eta}^{(m)} \in \boldsymbol{E}_M^m} \left( \prod_{i=1}^{m} \boldsymbol{x}_i \cdot \boldsymbol{\eta}_i \right) \boldsymbol{\eta}^{(m)}$$

$$\oplus \sum_{\boldsymbol{\eta}^{(m)} \in \boldsymbol{E}_M^m} \left( \prod_{i=1}^{m} \boldsymbol{x}_i \cdot \boldsymbol{\eta}_i \right) \sum_{j=1}^{M} (\boldsymbol{x}_{m+1} \cdot \boldsymbol{e}_j)(\boldsymbol{e}_j, )$$

$$= 1 \cdot \boldsymbol{x}^{(m)} \oplus 1 \cdot (\boldsymbol{x}_{m+1}, ) = \boldsymbol{x}^{(m+1)},$$

where in the penultimate equality we have used the inductive hypothesis and (twice) the result of the previous induction.

We can now prove the statement of the Lemma. Applying Jensen's inequality to equation (9) with the convex function $f$, we have that

$$f(\boldsymbol{x}_1, \ldots, \boldsymbol{x}_m) = f\left(\sum_{\boldsymbol{\eta}^{(m)} \in \boldsymbol{E}_M^m} \left(\prod_{i=1}^{m} \boldsymbol{x}_i \cdot \boldsymbol{\eta}_i\right) \boldsymbol{\eta}^{(m)}\right)$$

$$\leq \sum_{\boldsymbol{\eta}^{(m)} \in \boldsymbol{E}_M^m} \left(\prod_{i=1}^{m} \boldsymbol{x}_i \cdot \boldsymbol{\eta}_i\right) f\left(\boldsymbol{\eta}^{(m)}\right).$$

Let $\boldsymbol{\mu} = \mathbb{E}[\boldsymbol{X}_1]$ denote the mean of the i.i.d. random vectors $X_i$. Then the above inequality implies

$$\mathbb{E}[f(\boldsymbol{X}_1, \ldots, \boldsymbol{X}_m)] \leq \sum_{\boldsymbol{\eta}^{(m)} \in \boldsymbol{E}_M^m} \left(\prod_{i=1}^{m} \boldsymbol{\mu} \cdot \boldsymbol{\eta}_i\right) f\left(\boldsymbol{\eta}^{(m)}\right)$$

$$= \sum_{\boldsymbol{\eta}^{(m)} \in \boldsymbol{E}_M^m} \left(\prod_{i=1}^{m} \mathbb{P}(\boldsymbol{X}_i' = \boldsymbol{\eta}_i)\right) f\left(\boldsymbol{\eta}^{(m)}\right)$$

$$= \mathbb{E}[f(\boldsymbol{X}_1', \ldots, \boldsymbol{X}_m')].$$

## B.2  Proof of Lemma 8

The proof of Lemma 8 itself requires two technical helping lemmas which we now state and prove.

**Lemma 12.** *For any integers $n \geq 2$ and $p \geq -1$,*

$$\sum_{k=1}^{n-1} \frac{(n-k)^{p/2}}{\sqrt{k}} \leq n^{\frac{p+1}{2}} \int_0^1 \frac{(1-x)^{p/2}}{\sqrt{x}} dx.$$

*Proof.* The case of $p = -1$, namely

$$\sum_{k=1}^{n-1} \frac{1}{\sqrt{k(n-k)}} \leq \int_0^1 \frac{1}{\sqrt{x(1-x)}} dx,$$

has already been demonstrated in [22]. For $p > -1$, let

$$f_p(x) := \frac{(1-x)^{p/2}}{\sqrt{x}}.$$

We will show that each $f_p(\cdot)$ is monotonically decreasing on $(0, 1)$. Indeed,

$$\frac{df_p}{dx}(x) = -\frac{(1-x)^{\frac{p}{2}-1}(px + 1 - x)}{2x^{3/2}} \leq -\frac{(1-x)^{p/2}}{2x^{3/2}} < 0,$$

where for the inequalities we have used the fact that $p > -1$ and $x \in (0, 1)$. We therefore see that

$$\sum_{k=1}^{n-1} \frac{(n-k)^{p/2}}{\sqrt{k}} = \sum_{k=1}^{n-1} \frac{n^{p/2}(1 - \frac{k}{n})^{p/2}}{\sqrt{n}\sqrt{\frac{k}{n}}}$$

$$= n^{\frac{p+1}{2}} \sum_{k=1}^{n-1} \frac{1}{n} \frac{(1 - \frac{k}{n})^{p/2}}{\sqrt{\frac{k}{n}}}$$

$$= n^{\frac{p+1}{2}} \sum_{k=1}^{n-1} \frac{1}{n} f_p\left(\frac{k}{n}\right)$$

$$\leq n^{\frac{p+1}{2}} \sum_{k=1}^{n-1} \int_{\frac{k-1}{n}}^{\frac{k}{n}} f_p(x) dx$$

$$= n^{\frac{p+1}{2}} \int_0^{1-\frac{1}{n}} f_p(x)dx$$

$$\leq n^{\frac{p+1}{2}} \int_0^1 f_p(x)dx.$$

 $\qquad\square$

Intuitively, the proof of the above lemma works by bounding the integral below by a Riemann sum. In the following lemma we actually calculate this integral, yielding a more explicit bound on the sum in Lemma 12. We found it is easier to calculate a slightly more general integral, where the 1 in the limit and the integrand is replaced by a positive constant $a$.

**Lemma 13.** *For any real number $a > 0$ and integer $n \geq -1$,*

$$\int_0^a \frac{(a-x)^{n/2}}{\sqrt{x}}dx = \sqrt{\pi}\frac{\Gamma(\frac{n+2}{2})}{\Gamma(\frac{n+3}{2})}a^{\frac{n+1}{2}}.$$

*Proof.* Define

$$I_n(a) := \int_0^a \frac{(a-x)^{n/2}}{\sqrt{x}}dx \qquad \text{and} \qquad f_n(a) := \sqrt{\pi}\frac{\Gamma(\frac{n+2}{2})}{\Gamma(\frac{n+3}{2})}a^{\frac{n+1}{2}}.$$

We proceed by induction, increasing $n$ by 2 each time. This means we need two base cases. First, for $n = -1$, we have

$$I_{-1}(a) = \int_0^a \frac{1}{\sqrt{x(a-x)}}dx = \left[2\arcsin\sqrt{\frac{x}{a}}\right]_0^a = \pi = f_{-1}(a),$$

since $\Gamma(\frac{1}{2}) = \sqrt{\pi}$ and $\Gamma(1) = 1$. Second, for $n = 0$,

$$I_0(a) = \int_0^a \frac{1}{\sqrt{x}}dx = \left[2\sqrt{x}\right]_0^a = 2\sqrt{a} = f_0(a),$$

since $\Gamma(\frac{3}{2}) = \frac{\sqrt{\pi}}{2}$. Now, by the Leibniz integral rule, we have

$$\frac{d}{da}I_{n+2}(a) = \int_0^a \frac{\partial}{\partial a}\frac{(a-x)^{\frac{n+2}{2}}}{\sqrt{x}}dx = \frac{n+2}{2}\int_0^a \frac{(a-x)^{\frac{n}{2}}}{\sqrt{x}}dx = \frac{n+2}{2}I_n(a).$$

Thus

$$I_{n+2}(a) = \frac{n+2}{2}\left[\int_0^a I_n(t)dt + I_n(0)\right] = \frac{n+2}{2}\int_0^a I_n(t)dt,$$

since $I_n(0) = 0$.

Now, for the inductive step, suppose $I_n(a) = f_n(a)$ for some $n \geq -1$. Then, using the previous calculation, we have

$$I_{n+2}(a) = \frac{n+2}{2}\int_0^a f_n(t)dt$$

$$= \frac{n+2}{2}\int_0^a \sqrt{\pi}\frac{\Gamma(\frac{n+2}{2})}{\Gamma(\frac{n+3}{2})}t^{\frac{n+1}{2}}dt$$

$$= \sqrt{\pi}\frac{\frac{n+2}{2}\Gamma(\frac{n+2}{2})}{\frac{n+3}{2}\Gamma(\frac{n+3}{2})}a^{\frac{n+3}{2}}$$

$$= \sqrt{\pi}\frac{\Gamma(\frac{n+2}{2}+1)}{\Gamma(\frac{n+3}{2}+1)}a^{\frac{n+3}{2}}$$

$$= \sqrt{\pi}\frac{\Gamma\left(\frac{(n+2)+2}{2}\right)}{\Gamma\left(\frac{(n+2)+3}{2}\right)}a^{\frac{(n+2)+1}{2}}$$

$$= f_{n+2}(a).$$

This completes the proof. $\qquad\square$

We are now ready to prove Lemma 8 which, for ease of reference, we restate here. For integers $M \geq 1$ and $m \geq M$,

$$\sum_{\boldsymbol{k} \in S_{m,M}^{>0}} \frac{1}{\prod_{j=1}^{M} \sqrt{k_j}} \leq \frac{\pi^{\frac{M}{2}} m^{\frac{M-2}{2}}}{\Gamma(\frac{M}{2})}.$$

*Proof.* (of Lemma 8) We proceed by induction on $M$. For $M = 1$, the set $S_{m,M}$ contains a single element, namely the one-dimensional vector $\boldsymbol{k} = (k_1,) = (m,)$. In this case, the left hand side is $1/\sqrt{m}$ while the right hand side is $\sqrt{\pi}/(\sqrt{m}\Gamma(1/2)) = 1/\sqrt{m}$, since $\Gamma(1/2) = \sqrt{\pi}$.

Now, as the inductive hypothesis, assume the inequality of Lemma 8 holds for some fixed $M \geq 1$ and all $m \geq M$. Then for all $m \geq M + 1$, we have

$$\sum_{\boldsymbol{k} \in S_{m,M+1}^{>0}} \frac{1}{\prod_{j=1}^{M+1} \sqrt{k_j}} = \sum_{k_1=1}^{m-M} \frac{1}{\sqrt{k_1}} \sum_{\boldsymbol{k}' \in S_{m-k_1,M}^{>0}} \frac{1}{\prod_{j=1}^{M} \sqrt{k_j'}}$$

$$\leq \sum_{k_1=1}^{m-M} \frac{1}{\sqrt{k_1}} \frac{\pi^{\frac{M}{2}} (m-k_1)^{\frac{M-2}{2}}}{\Gamma(\frac{M}{2})} \qquad \text{(by the inductive hypothesis)}$$

$$= \frac{\pi^{\frac{M}{2}}}{\Gamma(\frac{M}{2})} \sum_{k_1=1}^{m-M} \frac{(m-k_1)^{\frac{M-2}{2}}}{\sqrt{k_1}}$$

$$\leq \frac{\pi^{\frac{M}{2}}}{\Gamma(\frac{M}{2})} \sum_{k_1=1}^{m-1} \frac{(m-k_1)^{\frac{M-2}{2}}}{\sqrt{k_1}} \qquad \text{(enlarging the sum domain)}$$

$$\leq \frac{\pi^{\frac{M}{2}}}{\Gamma(\frac{M}{2})} m^{\frac{M-1}{2}} \int_0^1 \frac{(1-x)^{\frac{M-2}{2}}}{\sqrt{x}} dx \qquad \text{(by Lemma 12)}$$

$$= \frac{\pi^{\frac{M}{2}}}{\Gamma(\frac{M}{2})} m^{\frac{M-1}{2}} \sqrt{\pi} \frac{\Gamma(\frac{M}{2})}{\Gamma(\frac{M+1}{2})} \qquad \text{(by Lemma 13)}$$

$$= \frac{\pi^{\frac{M+1}{2}} m^{\frac{M-1}{2}}}{\Gamma(\frac{M+1}{2})},$$

as required. $\qquad\qquad\qquad\qquad\qquad\qquad\qquad\qquad\qquad\qquad\qquad\qquad\qquad\qquad\qquad\qquad\qquad \square$

### B.3 Proof of Proposition 9

*Proof.* The case where $q_j = 1$ or $p_j = 1$ can be dealt with trivially by splitting into the three following subcases

- $q_j = p_j = 1 \implies \mathrm{kl}(q_j \| p_j) = \mathrm{kl}(\boldsymbol{q} \| \boldsymbol{p}) = 0$

- $q_j = 1, p_j \neq 1 \implies \mathrm{kl}(q_j \| p_j) = \mathrm{kl}(\boldsymbol{q} \| \boldsymbol{p}) = -\log p_j$

- $q_j \neq 1, p_j = 1 \implies \mathrm{kl}(q_j \| p_j) = \mathrm{kl}(\boldsymbol{q} \| \boldsymbol{p}) = \infty.$

For $q_j \neq 1$ and $p_j \neq 1$ define the distributions $\tilde{\boldsymbol{q}}, \tilde{\boldsymbol{p}} \in \triangle_M$ by $\tilde{q}_j = \tilde{p}_j = 0$ and

$$\tilde{q}_i = \frac{q_i}{1 - q_j} \qquad \text{and} \qquad \tilde{p}_i = \frac{p_i}{1 - p_j}$$

for $i \neq j$. Then

$$\sum_{i \neq j} q_i \log \frac{q_i}{p_i} = \sum_{i \neq j} (1 - q_j) \tilde{q}_i \log \frac{(1 - q_j) \tilde{q}_i}{(1 - p_j) \tilde{p}_i}$$

$$= (1 - q_j) \sum_{i \neq j} \tilde{q}_i \log \frac{\tilde{q}_i}{\tilde{p}_i} + \tilde{q}_i \log \frac{1 - q_j}{1 - p_j}$$

$$= (1 - q_j)\mathrm{kl}(\tilde{\boldsymbol{q}}\|\tilde{\boldsymbol{p}}) + (1 - q_j)\log\frac{1 - q_j}{1 - p_j}$$

$$\geq (1 - q_j)\log\frac{1 - q_j}{1 - p_j}.$$

The final inequality holds since $\mathrm{kl}(\tilde{\boldsymbol{q}}\|\tilde{\boldsymbol{p}}) \geq 0$. Further, note that we have equality if and only if $\tilde{\boldsymbol{q}} = \tilde{\boldsymbol{p}}$, which, by their definitions, translates to

$$p_i = \frac{1 - p_j}{1 - q_j}q_i$$

for all $i \neq j$. If we now add $q_j\log\frac{q_j}{p_j}$ to both sides, we obtain

$$\mathrm{kl}(\boldsymbol{q}\|\boldsymbol{p}) \geq (1 - q_j)\log\frac{1 - q_j}{1 - p_j} + q_j\log\frac{q_j}{p_j} = \mathrm{kl}(q_j\|p_j),$$

with the same condition for equality. $\qquad\square$

The following proposition makes more precise the argument found at the beginning of Section 4 for how Proposition 9 can be used to derive the tightest possible lower and upper bounds on each $R_D^j(Q)$.

**Proposition 14.** *Suppose that $\boldsymbol{q}, \boldsymbol{p} \in \triangle_M$ are such that $\mathrm{kl}(\boldsymbol{q}\|\boldsymbol{p}) \leq B$, where $\boldsymbol{q}$ is known and $\boldsymbol{p}$ is unknown. Then, in the absence of any further information, the tightest bound that can be obtained on each $p_j$ is*

$$p_j \leq \mathrm{kl}^{-1}(q_j, B).$$

*Proof.* Suppose $p_j > \mathrm{kl}^{-1}(q_j, B)$. Then, by definition of $\mathrm{kl}^{-1}$, we have that $\mathrm{kl}(q_j\|p_j) > B$. By Proposition 9, this would then imply $\mathrm{kl}(\boldsymbol{q}\|\boldsymbol{p}) > B$, contradicting our assumption. Therefore $p_j \leq \mathrm{kl}^{-1}(q_j, B)$. Now, with the information we have, we cannot rule out that

$$p_i = \frac{1 - p_j}{1 - q_j}q_i$$

for all $i \neq j$ and thus, by Proposition 9, that $\mathrm{kl}(q_j\|p_j) = \mathrm{kl}(\boldsymbol{q}\|\boldsymbol{p})$. Further, we cannot rule out that $\mathrm{kl}(\boldsymbol{q}\|\boldsymbol{p}) = B$. Thus, it is possible that $\mathrm{kl}(q_j\|p_j) = B$, in which case $p_j = \mathrm{kl}^{-1}(q_j, B)$. We therefore see that $\mathrm{kl}^{-1}(q_j, B)$ is the tightest possible upper bound on $p_j$, for each $j \in [M]$. $\qquad\square$

## B.4 Proof of Proposition 11

Before proving the proposition, we first argue that $\mathrm{kl}_{\boldsymbol{\ell}}^{-1}(\boldsymbol{u}|c)$ given by Definition 10 is well-defined. First, note that $A_{\boldsymbol{u}} := \{\boldsymbol{v} \in \triangle_M : \mathrm{kl}(\boldsymbol{u}\|\boldsymbol{v}) \leq c\}$ is compact (boundedness is clear and it is closed because it is the preimage of the closed set $[0, c]$ under the continuous map $\boldsymbol{v} \mapsto \mathrm{kl}(\boldsymbol{u}\|\boldsymbol{v})$) and so the continuous function $f_{\boldsymbol{\ell}}$ achieves its supremum on $A_{\boldsymbol{u}}$. Further, note that $A_{\boldsymbol{u}}$ is a convex subset of $\triangle_M$ (because the map $\boldsymbol{v} \mapsto \mathrm{kl}(\boldsymbol{u}\|\boldsymbol{v})$ is convex) and $f_{\boldsymbol{\ell}}$ is linear, so the supremum of $f_{\boldsymbol{\ell}}$ over $A_{\boldsymbol{u}}$ is achieved and is located on the boundary of $A_{\boldsymbol{u}}$. This means we can replace the inequality constraint $\mathrm{kl}(\boldsymbol{u}\|\boldsymbol{v}) \leq c$ in Definition 10 with the equality constraint $\mathrm{kl}(\boldsymbol{u}\|\boldsymbol{v}) = c$. Finally, if $\boldsymbol{u} \in \triangle_M^{>0}$ then $A_{\boldsymbol{u}}$ is a *strictly* convex subset of $\triangle_M$ (because the map $\boldsymbol{v} \mapsto \mathrm{kl}(\boldsymbol{u}\|\boldsymbol{v})$ is then *strictly* convex) and so the supremum of $f_{\boldsymbol{\ell}}$ occurs at a *unique* point on the boundary of $A_{\boldsymbol{u}}$. In other words, if $\boldsymbol{u} \in \triangle_M^{>0}$ then $\mathrm{kl}_{\boldsymbol{\ell}}^{-1}(\boldsymbol{u}|c)$ is defined *uniquely*.

*Proof.* (of Proposition 11) We start by deriving the implicit expression for $\boldsymbol{v}^*(\tilde{\boldsymbol{u}}) = \mathrm{kl}_{\boldsymbol{\ell}}^{-1}(\boldsymbol{u}|c)$ given in the proposition by solving a transformed version of the optimisation problem given by Definition 10 using the method of Lagrange multipliers. We obtain two solutions to the Lagrangian equations, which must correspond to the maximum and minimum total risk over the set $A_{\boldsymbol{u}} := \{\boldsymbol{v} \in \triangle_M : \mathrm{kl}(\boldsymbol{u}\|\boldsymbol{v}) \leq c\}$ because, as argued in the main text (see the discussion after Definition 10), $A_{\boldsymbol{u}}$ is compact and so the linear total risk $f_{\boldsymbol{\ell}}(\boldsymbol{v})$ attains its maximum and minimum on $A_{\boldsymbol{u}}$.

By definition of $\boldsymbol{v}^*(\tilde{\boldsymbol{u}}) = \mathrm{kl}_{\boldsymbol{\ell}}^{-1}(\boldsymbol{u}|c)$, we know that $\mathrm{kl}(\boldsymbol{v}^*(\tilde{\boldsymbol{u}})\|\boldsymbol{u}) \leq c$. Since, by assumption, $u_j > 0$ for all $j$, we see that $\boldsymbol{v}^*(\tilde{\boldsymbol{u}})_j > 0$ for all $j$, otherwise we would have $\mathrm{kl}(\boldsymbol{v}^*(\tilde{\boldsymbol{u}})\|\boldsymbol{u}) = \infty$, a

651 contradiction. Thus $\boldsymbol{v}^*(\tilde{\boldsymbol{u}}) \in \triangle_M^{>0}$ and we are permitted to instead optimise over the unbounded
652 variable $\boldsymbol{t} \in \mathbb{R}^M$, where $t_j := \ln v_j$. With this transformation, the constraint $\boldsymbol{v} \in \triangle_M$ can be
653 replaced simply by $\sum_j e^{t_j} = 1$ and the optimisation problem becomes

$$\text{Maximise:} \quad F(\boldsymbol{t}) := \sum_{j=1}^{M} \ell_j e^{t_j}$$

$$\text{Subject to:} \quad g(\boldsymbol{t}; \boldsymbol{u}, c) := \mathrm{kl}(\boldsymbol{u} \| e^{\boldsymbol{t}}) - c = 0,$$

$$h(\boldsymbol{t}) := \sum_{j=1}^{M} e^{t_j} - 1 = 0,$$

654 where $e^{\boldsymbol{t}} \in \mathbb{R}^M$ is defined by $(e^{\boldsymbol{t}})_j := e^{t_j}$. Note that $F(\boldsymbol{t}) = f_\ell(e^{\boldsymbol{t}})$. Following the terminology
655 of mathematical economics, we call the $t_j$ the *optimisation variables*, and the $\tilde{u}_j$ (namely the $u_j$
656 and $c$) the *choice variables*. The vector $\ell$ is considered fixed—we neither want to optimise over
657 it nor differentiate with respect to it—which is why we occasionally suppress it from the notation
658 henceforth.

659 For each $\tilde{\boldsymbol{u}}$, let $\boldsymbol{v}^*(\tilde{\boldsymbol{u}})$ and $\boldsymbol{t}^*(\tilde{\boldsymbol{u}})$ be the solutions to the original and transformed optimisation
660 problems respectively. Since the map $\boldsymbol{v} = e^{\boldsymbol{t}}$ is one-to-one, it is clear that since $\boldsymbol{v}^*(\tilde{\boldsymbol{u}})$ exists uniquely,
661 so does $\boldsymbol{t}^*(\tilde{\boldsymbol{u}})$, and that they are related by $\boldsymbol{v}^*(\tilde{\boldsymbol{u}}) = e^{\boldsymbol{t}^*(\tilde{\boldsymbol{u}})}$. We therefore have the identity

$$f_\ell(\boldsymbol{v}^*(\tilde{\boldsymbol{u}})) \equiv F(\boldsymbol{t}^*(\tilde{\boldsymbol{u}})).$$

662 Recalling that $f_\ell^*(\tilde{\boldsymbol{u}}) := f_\ell(\boldsymbol{v}^*(\tilde{\boldsymbol{u}}))$, we see that

$$\nabla_{\tilde{\boldsymbol{u}}} f_\ell^*(\tilde{\boldsymbol{u}}) \equiv \nabla_{\tilde{\boldsymbol{u}}} F(\boldsymbol{t}^*(\tilde{\boldsymbol{u}})). \tag{10}$$

663 the derivatives of $f_\ell(\mathrm{kl}_\ell^{-1}(\boldsymbol{u}|c))$ with respect to $\boldsymbol{u}$ and $c$ are given by $\nabla_{\tilde{\boldsymbol{u}}} F(\boldsymbol{t}^*(\tilde{\boldsymbol{u}}))$.

664 Using the method of Lagrange multipliers, there exist real numbers $\lambda^* = \lambda^*(\tilde{\boldsymbol{u}})$ and $\mu^* = \mu^*(\tilde{\boldsymbol{u}})$
665 such that $(\boldsymbol{t}^*, \lambda^*, \mu^*)$ is a stationary point (with respect to $\boldsymbol{t}, \lambda$ and $\mu$) of the Lagrangian function

$$\mathcal{L}(\boldsymbol{t}, \lambda, \mu; \tilde{\boldsymbol{u}}) := F(\boldsymbol{t}) + \lambda g(\boldsymbol{t}; \tilde{\boldsymbol{u}}) + \mu h(\boldsymbol{t}).$$

666 Let $F_{\boldsymbol{t}}(\cdot)$ and $h_{\boldsymbol{t}}(\cdot)$ denote the gradient vectors of $F$ and $h$ respectively, and let $g_{\boldsymbol{t}}(\cdot\,; \tilde{\boldsymbol{u}})$ and $g_{\tilde{\boldsymbol{u}}}(\boldsymbol{t}; \cdot\,)$
667 denote the gradient vectors of $g$ with respect to $\boldsymbol{t}$ only and $\tilde{\boldsymbol{u}}$ only, respectively. Simple calculation
668 yields

$$g_{\boldsymbol{t}}(\boldsymbol{t}; \tilde{\boldsymbol{u}}) = \left( \frac{\partial g}{\partial t_1}(\boldsymbol{t}; \tilde{\boldsymbol{u}}), \dots, \frac{\partial g}{\partial t_M}(\boldsymbol{t}; \tilde{\boldsymbol{u}}) \right) = -\boldsymbol{u} \quad \text{and}$$

$$g_{\tilde{\boldsymbol{u}}}(\boldsymbol{t}; \tilde{\boldsymbol{u}}) = \left( \frac{\partial g}{\partial \tilde{u}_1}(\boldsymbol{t}; \tilde{\boldsymbol{u}}), \dots, \frac{\partial g}{\partial \tilde{u}_{M+1}}(\boldsymbol{t}; \tilde{\boldsymbol{u}}) \right) = \left( 1 - t_1 + \log u_1, \dots, 1 - t_M + \log u_M, -1 \right). \tag{11}$$

669 Then, taking the partial derivatives of $\mathcal{L}$ with respect to $\lambda, \mu$ and the $t_j$, we have that $(\boldsymbol{t}, \lambda, \mu) =$
670 $(\boldsymbol{t}^*(\tilde{\boldsymbol{u}}), \lambda^*(\tilde{\boldsymbol{u}}), \mu^*(\tilde{\boldsymbol{u}}))$ solves the simultaneous equations

$$F_{\boldsymbol{t}}(\boldsymbol{t}) + \lambda g_{\boldsymbol{t}}(\boldsymbol{t}; \tilde{\boldsymbol{u}}) + \mu h_{\boldsymbol{t}}(\boldsymbol{t}) = \boldsymbol{0}, \tag{12}$$

671

$$g(\boldsymbol{t}; \tilde{\boldsymbol{u}}) = 0, \quad \text{and}$$
672
$$h(\boldsymbol{t}) = 0,$$

673 where the last two equations recover the constraints. Substituting the gradients $F_{\boldsymbol{t}}, g_{\boldsymbol{t}}$ and $h_{\boldsymbol{t}}$, the first
674 equation reduces to

$$\ell \odot e^{\boldsymbol{t}} - \lambda \boldsymbol{u} + \mu e^{\boldsymbol{t}} = \boldsymbol{0},$$

675 which implies that for all $j \in [M]$

$$e^{t_j} = \frac{\lambda u_j}{\mu + \ell_j}. \tag{13}$$

676 Substituting this into the constraints $g = h = 0$ yields the following simultaneous equations in $\lambda$ and
677 $\mu$

$$c = \mathrm{kl}(\boldsymbol{u} \| e^{\boldsymbol{t}}) = \sum_{j=1}^{M} u_j \log \frac{u_j}{e^{t_j}} = \sum_{j=1}^{M} u_j \log \frac{\mu + \ell_j}{\lambda} \quad \text{and} \quad \lambda \sum_{j=1}^{M} \frac{u_j}{\mu + \ell_j} = 1.$$

Substituting the second into the first and rearranging the second, this is equivalent to solving

$$c = \sum_{j=1}^{M} u_j \log\left((\mu + \ell_j)\sum_{k=1}^{M}\frac{u_k}{\mu + \ell_k}\right) \quad \text{and} \quad \lambda = \left(\sum_{j=1}^{M}\frac{u_j}{\mu + \ell_j}\right)^{-1}. \tag{14}$$

It has already been established in the discussion after Definition 10 that $f_{\boldsymbol{\ell}}(\boldsymbol{v})$ attains its maximum on the set $A_{\boldsymbol{u}} := \{\boldsymbol{v} \in \triangle_M : \mathrm{kl}(\boldsymbol{u}\|\boldsymbol{v}) \leq c\}$. Therefore $F(\boldsymbol{t})$ also attains its maximum on $\mathbb{R}^M$ and one of the solutions to these simultaneous equations corresponds to this maximum. We first show that there is a single solution to the first equation in the set $(-\infty, -\max_j \ell_j)$, referred to as $\mu^*(\tilde{\boldsymbol{u}})$ in the proposition. Second, we show that any other solution corresponds to a smaller total risk, so that $\mu^*(\tilde{\boldsymbol{u}})$ corresponds to the maximum total risk and yields $\boldsymbol{v}^*(\tilde{\boldsymbol{u}}) = \mathrm{kl}_{\boldsymbol{\ell}}^{-1}(\boldsymbol{u}|c)$ when $\mu^*(\tilde{\boldsymbol{u}})$ and the associated $\lambda^*(\tilde{\boldsymbol{u}})$ are substituted into Equation 13.

For the first step, note that since the $e^{t_j}$ are probabilities, we see from Equation 13 that either $\mu + \ell_j > 0$ for all $j$ (in the case that $\lambda > 0$), or $\mu + \ell_j < 0$ for all $j$ (in the case that $\lambda < 0$). Thus any solutions $\mu$ to the first equation must be in $(-\infty, -\max_j \ell_j)$ or $(-\min_j \ell_j, \infty)$. If $\mu \in (-\infty, -\max_j \ell_j)$ then the first equation can be written as $c = \phi_{\boldsymbol{\ell}}(\mu)$, with $\phi_{\boldsymbol{\ell}}$ as defined in the statement of the proposition. We now show that $\phi_{\boldsymbol{\ell}}$ is strictly increasing in $\mu$, and that $\phi_{\boldsymbol{\ell}}(\mu) \to 0$ as $\mu \to -\infty$ and $\phi_{\boldsymbol{\ell}}(\mu) \to \infty$ as $\mu \to -\max_j \ell_j$, so that $c = \phi_{\boldsymbol{\ell}}(\mu)$ does indeed have a single solution in the set $(-\infty, -\max_j \ell_j)$. Straightforward differentiation and algebra shows that

$$\phi_{\boldsymbol{\ell}}'(\mu) = \sum_{j=1}^{M}\frac{u_j}{(\mu + \ell_j)\sum_{k=1}^{M}\frac{u_k}{\mu+\ell_k}}\left(\sum_{k'=1}^{M}\frac{u_{k'}}{\mu + \ell_{k'}} - (\mu + \ell_j)\sum_{k'=1}^{M}\frac{u_{k'}}{(\mu + \ell_{k'})^2}\right)$$

$$= \frac{\left(\sum_{j=1}^{M}\frac{u_j}{\mu+\ell_j}\right)^2 - \sum_{j=1}^{M}\frac{u_j}{(\mu+\ell_j)^2}}{\sum_{k=1}^{M}\frac{u_k}{\mu+\ell_k}}.$$

Jensen's inequality demonstrates that the numerator is strictly negative, where strictness is due to the assumption that the $\ell_j$ are not all equal. Further, since the denominator is strictly negative (since we are dealing with the case where $\mu \in (-\infty, -\max_j \ell_j)$), we see that $\phi_{\boldsymbol{\ell}}$ is strictly increasing for $\mu \in (-\infty, -\max_j \ell_j)$.[2] Turning to the limits, we first show that $\phi_{\boldsymbol{\ell}}(\mu) \to \infty$ as $\mu \to -\max_j \ell_j$.

We now determine the left hand limit. Define $J = \{j \in [M] : \ell_j = \max_k \ell_k\}$, noting that this is a strict subset of $[M]$ since by assumption the $\ell_j$ are not all equal. We then have that for $\mu \in (-\infty, \max_j \ell_j)$

$$e^{\phi_{\boldsymbol{\ell}}(\mu)} = \left(-\sum_{j=1}^{M}\frac{u_j}{\mu + \ell_j}\right)\left(\prod_{k=1}^{M}\left(-(\mu + \ell_k)\right)^{u_k}\right)$$

$$= \left(-\sum_{j\in J}\frac{u_j}{\mu + \ell_j} - \sum_{j'\notin J}\frac{u_{j'}}{\mu + \ell_{j'}}\right)\prod_{k\in J}\left(-(\mu + \ell_k)\right)^{u_k}\prod_{k'\notin J}\left(-(\mu + \ell_{k'})\right)^{u_{k'}}$$

$$\geq \left(-\sum_{j\in J}\frac{u_j}{\mu + \ell_j}\right)\prod_{k\in J}\left(-(\mu + \ell_k)\right)^{u_k}\prod_{k'\notin J}\left(-(\mu + \ell_{k'})\right)^{u_{k'}}$$

$$= \frac{\left(\sum_{j\in J}u_j\right)\left(\prod_{k'\notin J}\left(-(\mu + \ell_{k'})\right)^{u_{k'}}\right)}{\left(-(\mu + \max_j \ell_j)\right)^{1-\sum_{k\in J}u_k}}.$$

The first term in the numerator is a positive constant, independent of $\mu$. The second term in the numerator tends to a finite positive limit as $\mu \uparrow -\max_j \ell_j$. Since $[M]\setminus J$ is non-empty, the power in the denominator is positive and the term in the outer brackets is positive and tends to zero as $\mu \uparrow -\max_j \ell_j$. Thus $e^{\phi_{\boldsymbol{\ell}}(\mu)} \to \infty$ as $\mu \uparrow -\max_j \ell_j$ and, by the continuity of the logarithm, $\phi_{\boldsymbol{\ell}}(\mu)$ as $\mu \uparrow -\max_j \ell_j$.

---

[2]Incidentally, this argument also shows that there is at most one solution to the first equation in (14) in the range $(-\min_j \ell_j, \infty)$. There indeed exists a unique solution, which corresponds to the minimum total risk, but we do not prove this.

We now determine $\lim_{\mu \to -\infty} \phi_{\boldsymbol{\ell}}(\mu)$ by sandwiching $\phi(\mu)$ between two functions that both tend to zero as $\mu \to -\infty$. First, since $\ell_j \geq 0$ for all $j$, for $\mu \in (-\infty, -\max_j \ell_j)$ we have

$$\log\left(-\sum_{j=1}^{M} \frac{u_j}{\mu + \ell_j}\right) \geq \log\left(-\sum_{j=1}^{M} \frac{u_j}{\mu}\right) = -\log(-\mu) = -\sum_{j=1}^{M} u_j \log(-\mu),$$

and so

$$\phi_{\boldsymbol{\ell}}(\mu) \geq -\sum_{j=1}^{M} u_j \log(-\mu) + \sum_{j=1}^{M} u_j \log\left(-(\mu + \ell_j)\right) = \sum_{j=1}^{M} u_j \log\left(1 + \frac{\ell_j}{\mu}\right) \to 0 \quad \text{as} \quad \mu \to -\infty.$$

Similarly,

$$\sum_{j=1}^{M} u_j \log\left(-(\mu + \ell_j)\right) \leq \sum_{j=1}^{M} u_j \log(-\mu) = \log(-\mu),$$

and so

$$\phi_{\boldsymbol{\ell}}(\mu) \leq \log\left(\mu \sum_{j=1}^{M} \frac{u_j}{\mu + \ell_j}\right) = \log\left(\sum_{j=1}^{M} \frac{u_j}{1 + \frac{\ell_j}{\mu}}\right) \to 0 \quad \text{as} \quad \mu \to -\infty.$$

This completes the first step, namely showing that there does indeed exist a unique solution $\mu^*(\tilde{\boldsymbol{u}})$ in the set $(-\ell_1, \infty)$ to the first equation in line (14).

We now turn to the second step, namely showing that this solution corresponds to the maximum total risk. Given a value of the Lagrange multiplier $\mu$, substitution into Equation 13 gives

$$e^{t_j}(\mu) = \frac{\frac{u_j}{\mu + \ell_j}}{\sum_{k=1}^{M} \frac{u_k}{\mu + \ell_k}}$$

and therefore total risk

$$R(\mu) = \frac{\sum_{j=1}^{M} \frac{u_j \ell_j}{\mu + \ell_j}}{\sum_{k=1}^{M} \frac{u_k}{\mu + \ell_k}}.$$

To prove that the solution $\mu^*(\tilde{\boldsymbol{u}}) \in (-\infty, -\max_j \ell_j)$ is the solution to the first equation in line (14) that maximises $R$, it suffices to show that $R(\mu) \to \sum_{j=1}^{M} u_j \ell_j$ as $|\mu| \to \infty$ and $R'(\mu) \geq 0$ for all $\mu \in (-\infty, -\max_j \ell_j) \cup (-\min_j \ell_j, \infty)$, so that

$$\inf_{\mu \in (-\infty, -\max_j \ell_j)} R(\mu) \geq \sup_{\mu \in (-\min_j \ell_j, \infty)} R(\mu).$$

This suffices as we have already proved that $\mu^*(\tilde{\boldsymbol{u}})$ is the only solution in $(-\infty, -\max_j \ell_j)$ to the first equation in line (14), and that no solutions exists in the set $[-\max_j \ell_j, -\min_j \ell_j]$.

The limit can be easily evaluated by first rewriting $R(\mu)$ and then taking the limit as $|\mu| \to \infty$ as follows

$$R(\mu) = \frac{\sum_{j=1}^{M} \frac{u_j \ell_j}{1 + \frac{\ell_j}{\mu}}}{\sum_{k=1}^{M} \frac{u_k}{1 + \frac{\ell_k}{\mu}}} \to \frac{\sum_{j=1}^{M} u_j \ell_j}{\sum_{k=1}^{M} u_k} = \sum_{j=1}^{M} u_j \ell_j.$$

To show that $R'(\mu) \geq 0$, let $\ell_{(j)}$ denote the $j$'th smallest component of $\boldsymbol{\ell}$ (breaking ties arbitrarily), so that $\ell_{(1)} \leq \cdots \leq \ell_{(M)}$, and use the quotient rule to see that

$$R'(\mu) \geq 0 \iff \frac{\left(\sum_{k=1}^{M} \frac{u_k}{\mu + \ell_k}\right)\left(\sum_{j=1}^{M} \frac{-u_j \ell_j}{(\mu + \ell_j)^2}\right) - \left(\sum_{j=1}^{M} \frac{u_j \ell_j}{\mu + \ell_j}\right)\left(\sum_{k=1}^{M} \frac{-u_k}{(\mu + \ell_k)^2}\right)}{\left(\sum_{p=1}^{M} \frac{u_p}{\mu + \ell_p}\right)^2} \geq 0$$

$$\iff \sum_{j=1}^{M}\sum_{k=1}^{M} \frac{u_j u_k \ell_j}{(\mu + \ell_j)(\mu + \ell_k)}\left(\frac{1}{\mu + \ell_k} - \frac{1}{\mu + \ell_j}\right) \geq 0$$

$$\iff \sum_{\substack{j,k\in[M]\\k<j}} \frac{u_j u_k \ell_{(j)}}{(\mu+\ell_{(j)})(\mu+\ell_{(k)})}\left(\frac{1}{\mu+\ell_{(k)}}-\frac{1}{\mu+\ell_{(j)}}\right)$$

$$+\sum_{\substack{j,k\in[M]\\k>j}} \frac{u_j u_k \ell_{(j)}}{(\mu+\ell_{(j)})(\mu+\ell_{(k)})}\left(\frac{1}{\mu+\ell_{(k)}}-\frac{1}{\mu+\ell_{(j)}}\right)\geq 0,$$

where in the final line we have dropped the summands where $k=j$ since they equal zero as the terms in the bracket cancel. This final inequality holds since the first sum can be bounded below by the negative of the second sum as follows

$$\sum_{\substack{j,k\in[M]\\k<j}} \frac{u_j u_k \ell_{(j)}}{(\mu+\ell_{(j)})(\mu+\ell_{(k)})}\left(\frac{1}{\mu+\ell_{(k)}}-\frac{1}{\mu+\ell_{(j)}}\right)$$

$$\geq \sum_{\substack{j,k\in[M]\\k<j}} \frac{u_j u_k \ell_{(k)}}{(\mu+\ell_{(j)})(\mu+\ell_{(k)})}\left(\frac{1}{\mu+\ell_{(k)}}-\frac{1}{\mu+\ell_{(j)}}\right) \quad \text{(since } \ell_{(k)}\leq\ell_{(j)} \text{ for } k<j)$$

$$= \sum_{\substack{j,k\in[M]\\k>j}} \frac{u_k u_j \ell_{(j)}}{(\mu+\ell_{(k)})(\mu+\ell_{(j)})}\left(\frac{1}{\mu+\ell_{(j)}}-\frac{1}{\mu+\ell_{(k)}}\right) \quad \text{(swapping dummy variables } j,k).$$

We now turn to finding the partial derivatives of $F(\boldsymbol{t}^*(\tilde{\boldsymbol{u}}))$ with respect the $\tilde{u}_j$, which in turn will allow us to find the partial derivatives of $\mathrm{kl}_{\boldsymbol{\ell}}^{-1}(\boldsymbol{u}|c)$. Let $\nabla_{\tilde{\boldsymbol{u}}}$ denote the gradient operator with respect to $\tilde{\boldsymbol{u}}$. Then the quantity we are after is $\nabla_{\tilde{\boldsymbol{u}}}F(\boldsymbol{t}^*(\tilde{\boldsymbol{u}}))\in\mathbb{R}^{M+1}$, the $j$'th component of which is

$$\left(\nabla_{\tilde{\boldsymbol{u}}}F(\boldsymbol{t}^*(\tilde{\boldsymbol{u}}))\right)_j = \sum_{k=1}^{M+1}\frac{\partial F}{\partial t_k}(\boldsymbol{t}^*(\tilde{\boldsymbol{u}}))\frac{\partial t_k^*}{\partial \tilde{u}_j}(\tilde{\boldsymbol{u}}) = F_{\boldsymbol{t}}(\boldsymbol{t}^*(\tilde{\boldsymbol{u}}))\cdot\frac{\partial \boldsymbol{t}^*}{\partial \tilde{u}_j}(\tilde{\boldsymbol{u})}\in\mathbb{R}.$$

Thus the full gradient vector is

$$\nabla_{\tilde{\boldsymbol{u}}}F(\boldsymbol{t}^*(\tilde{\boldsymbol{u}})) = F_{\boldsymbol{t}}(\boldsymbol{t}^*(\tilde{\boldsymbol{u}}))\nabla_{\tilde{\boldsymbol{u}}}\boldsymbol{t}^*(\tilde{\boldsymbol{u}}), \tag{15}$$

where $\nabla_{\tilde{\boldsymbol{u}}}\boldsymbol{t}^*(\tilde{\boldsymbol{u}})$ is the $M\times(M+1)$ matrix given by

$$\left(\nabla_{\tilde{\boldsymbol{u}}}\boldsymbol{t}^*(\tilde{\boldsymbol{u}})\right)_{j,k} = \frac{\partial t_k^*}{\partial \tilde{u}_j}(\tilde{\boldsymbol{u}}).$$

Finding an expression for this matrix is difficult. Fortunately we can avoid needing to by using a trick from mathematical economics referred to as the envelope theorem, as we now show.

First, note that since, for all $\tilde{\boldsymbol{u}}$, the constraints $g=h=0$ are satisfied by $\boldsymbol{t}^*(\tilde{\boldsymbol{u}})$, we have the identities

$$g(\boldsymbol{t}^*(\tilde{\boldsymbol{u}}),\tilde{\boldsymbol{u}})\equiv 0 \quad \text{and} \quad h(\boldsymbol{t}^*(\tilde{\boldsymbol{u}}))\equiv 0.$$

Differentiating these identities with respect to $\tilde{u}_j$ then yields

$$g_{\boldsymbol{t}}(\boldsymbol{t}^*(\tilde{\boldsymbol{u}}),\tilde{\boldsymbol{u}})\cdot\frac{\partial \boldsymbol{t}^*}{\partial \tilde{u}_j}(\tilde{\boldsymbol{u}}) + g_{\tilde{u}_j}(\boldsymbol{t}^*(\tilde{\boldsymbol{u}}),\tilde{\boldsymbol{u}})\equiv 0 \quad \text{and} \quad h_{\boldsymbol{t}}(\boldsymbol{t}^*(\tilde{\boldsymbol{u}}))\cdot\frac{\partial \boldsymbol{t}^*}{\partial \tilde{u}_j}(\tilde{\boldsymbol{u}})\equiv 0.$$

As before, we can write these $M+1$ pairs of equations as the following pair of matrix equations

$$g_{\boldsymbol{t}}(\boldsymbol{t}^*(\tilde{\boldsymbol{u}}),\tilde{\boldsymbol{u}})\nabla_{\tilde{\boldsymbol{u}}}\boldsymbol{t}^*(\tilde{\boldsymbol{u}}) + g_{\tilde{\boldsymbol{u}}}(\boldsymbol{t}^*(\tilde{\boldsymbol{u}}),\tilde{\boldsymbol{u}})\equiv \boldsymbol{0} \quad \text{and} \quad h_{\boldsymbol{t}}(\boldsymbol{t}^*(\tilde{\boldsymbol{u}}))\nabla_{\tilde{\boldsymbol{u}}}\boldsymbol{t}^*(\tilde{\boldsymbol{u}})\equiv \boldsymbol{0}.$$

Multiplying these identities by $\lambda^*(\tilde{\boldsymbol{u}})$ and $\mu^*(\tilde{\boldsymbol{u}})$ respectively, and combining with equation (15), yields

$$\nabla_{\tilde{\boldsymbol{u}}}F(\boldsymbol{t}^*(\tilde{\boldsymbol{u}})) = \Big(F_{\boldsymbol{t}}(\boldsymbol{t}^*(\tilde{\boldsymbol{u}})) + \lambda^*(\tilde{\boldsymbol{u}})g_{\boldsymbol{t}}(\boldsymbol{t}^*(\tilde{\boldsymbol{u}}),\tilde{\boldsymbol{u}}) + \mu^*(\tilde{\boldsymbol{u}})h_{\boldsymbol{t}}(\boldsymbol{t}^*(\tilde{\boldsymbol{u}}))\Big)\nabla_{\tilde{\boldsymbol{u}}}\boldsymbol{t}^*(\tilde{\boldsymbol{u}})$$

$$+ \lambda^*(\tilde{\boldsymbol{u}})g_{\tilde{\boldsymbol{u}}}(\boldsymbol{t}^*(\tilde{\boldsymbol{u}}),\tilde{\boldsymbol{u}})$$

$$= \lambda^*(\tilde{\boldsymbol{u}})g_{\tilde{\boldsymbol{u}}}(\boldsymbol{t}^*(\tilde{\boldsymbol{u}}),\tilde{\boldsymbol{u}}),$$

where the final equality comes from noting that the terms in the large bracket vanish due to equation (12). Recalling the expression for $g_{\tilde{u}}(t; \tilde{u})$ given by Equation 11 and that $v^*(\tilde{u}) = \exp(t^*(\tilde{u}))$ we obtain

$$\nabla_{\tilde{u}} F(t^*(\tilde{u})) = \lambda^*(\tilde{u})\Big(1 - t^*(\tilde{u})_1 + \log u_1, \ldots, 1 - t^*(\tilde{u})_M + \log u_M, -1\Big)$$

$$= \lambda^*(\tilde{u})\left(1 + \log \frac{u_1}{v^*(\tilde{u})_1}, \ldots, 1 + \log \frac{u_M}{v^*(\tilde{u})_M}, -1\right)$$

Finally, recalling Equivalence (10), namely $\nabla_{\tilde{u}} f_\ell^*(\tilde{u}) \equiv \nabla_{\tilde{u}} F(t^*(\tilde{u}))$, we see that the above expression gives the derivatives $\frac{\partial f_\ell^*}{\partial u_j}(\tilde{u})$ and $\frac{\partial f_\ell^*}{\partial c}(\tilde{u})$ stated in the proposition, thus completing the proof. □