# OpenReview forum: "Controlling Confusion via Generalisation Bounds"
_NeurIPS.cc/2022/Conference — NeurIPS 2022 Submitted_

### Official Review · Reviewer_pSUq · 2022-07-09

**Rating:** 7
**Confidence:** 2
**Soundness:** 4 excellent
**Presentation:** 2 fair
**Contribution:** 3 good

**Summary:**

The paper deals with multi-class classification with a vector of $M$ possible error types. The paper establishes a PAC-Bayes bound for the convergence of the empirical error vector to the expected one,  under some convex distance measure (Thm. 3), and gives a more explicit bound for the kl-measure between the Multininomial distributions parametrized by those two corresponding vectors (Cor. 7). Finally, the paper discusses how the result of Cor.7 can yield a bound for the individual error types, and how to derive a gradient-based algorithm that minimizes the bound.


**Questions:**

1.
Line 170 -  I didn’t understand the comment about $m’$. How is the formulation with $\beta$ more general?


**Limitations:**

No further limitations

**Strengths And Weaknesses:**

### Strengths

1. The problem of multi-class generalization bounds for the confusion matrix is interesting.
2. The main results (Thm. 3, Cor. 7) seem elegant and non-trivial to derive.

### Weaknesses

1. The writing can be improved - for example - paragraph 38-48 is hard to follow, The structure of the paper can be improved, currently, the main contributions are hard to discern from smaller details.

2. Classical PAC-Bayes bounds (e.g. Maurer ‘04) can account for non-zero-one losses.  An alternative route to deriving bounds on the divergence of the empirical confusion matrix from the expected one, under some matrix norm, can be using those classical bounds for each error type and then using the union bound. How does this simple approach compares to the results in the paper?  I think the paper lacks comparisons to possible simpler approaches.

3. In the introduction, the authors describe a problem where each error type is associated with a different loss value $\ell_j \in [0, \infty)$. However, if am not mistaken, the bound presented in Thm 3. and Cor. 7 seem to only deal with $\ell_j \in {0,1}$.

4. There are also no explicit results for the convergence of the confusion matrix under some matrix norms.
I may have not understood correctly the results, in that case, I suggest the authors improve the clarity of these issues, by emphasizing the corresponding formulations and results.

5. There is no experimental validation for the suggested algorithm and for the bounds.

---

> ### Author Response · Authors · 2022-08-01
> **Many thanks for taking the time to give a detailed review of our work. We are especially encouraged by your acknowledgement that our results were non-trivial to derive and by your overall positive evaluation of our submission**
>
> We are glad you appreciate both our setting and our theoretical contribution to its analysis.
>
> We note you consider the paper “hard to follow” and believe the writing and structure “can be improved”. Since the other two reviewers found the paper to be “clearly written” (reviewer x9gX) and “well presented and easy to understand” (reviewer 9zyv), we would be especially grateful if you would kindly point to concrete issue, which we would then consider incorporating into a revised version. We agree the main results could be more prominently emphasised, and will revise the paper accordingly.
>
> Existing PAC-Bayes bounds can indeed be used to bound weightings of error types, and yes, a union bound can be used to bound several weightings simultaneously (albeit with a weakening of the bound). Using our bound however, no union bound is required – bounding the kl provides bounds on all uncountably many weightings, all of which hold simultaneously with high probability. Previous methods cannot achieve this.
>
> Let us explain this in more detail. Hitherto, PAC-Bayes bounds have been restricted to bounding one-dimensional measures of risk – the probability of error or the expected loss. We feel this is an impoverished view, as will illustrate in the case of classifying a medical image as containing either a benign tumour or a potentially malignant one requiring a biopsy. Then the balance of type one and type two errors must be finely controlled. Yes ordinary PAC-Bayes can account for this by weighting the different error types and then bounding this one-dimensional metric, but if it is later found that biopsies are riskier than expected, the original PAC-Bayes bound becomes useless and retraining is required to account for the new weighting. One can bound multiple weightings simultaneously using a union bound (at the expense of a less tight bound), but the significant advantage of our method is that it bounds *all uncountably many weightings simultaneously*, requiring neither a union bound nor retraining. A far richer measure of generalisation is thus provided – one obtains a bound on the type one error, the type two error, and every possible weighting of the two, all of which hold with high probability simultaneously. New knowledge of the risks of biopsies could be seamlessly integrated.
>
> We now address your comments on the loss. As explained in lines 73-76 of the introduction, the empirical and expected risk vectors are simply the empirical and expected counts of the number of errors of each type. No loss (0-1 or otherwise) is used at this stage, only the frequency of the different error types is being tracked. This is why Theorem 3 and Corollary 7 make no mention of a loss function – indeed, none is being used! It is only later, in Section 4 (line 308 onwards), where we pick up on the second step outlined lines 80-84 of the introduction, namely associating to each error type a loss value $\ell_j$ and bounding the corresponding total risk, defined on line 156. In short, Theorem 3 and Corollary 7 concern only the *frequency* of the different error types. Only once a user-specified loss vector $\ell$ has been decided does one bound a particular weighting of the error types using the inverse of our extension of the kl.
>
> As to your comments on convergence under a matrix norm, this is indeed an approach that has been taken in previous papers – see our citations [27] (“PAC-Bayesian generalization bound on confusion matrix for multi-class classification”, Morvant et al., 2012) and [17] (“On multi-class classification through the minimization of the confusion matrix norm”, Koço et al., 2013) discussed in lines 26-32. This approach is very different from ours however; ours is more flexible as it permits user-specified error types. While these error types can correspond to all the cells in a confusion matrix, the user has the additional freedom to only separate relevantly different errors in the hopes of obtaining a tighter bound. Thus one may not even be interested in the convergence behaviour of the full confusion matrix, since the full matrix is not being considered.
>
> While we did not include experiments, we feel that our contributions are mostly theoretical in nature and significant in their own right, and that an extensive numerical study would be a distinct contribution. Nevertheless, one can be reassured that our suggested algorithm (Appendix A) closely follows the usual approach to implementing PAC-Bayes bounds found and in the literature (for example, see “Computing nonvacuous generalization bounds for deep (stochastic) neural networks with many more parameters than training data”, Dziugaite and Roy, 2017). We only require the additional step of coding the derivatives of kl^-1, which can be obtained using Proposition 11 (which, despite appearances, is straightforward to implement).
>
> Finally, our formulation is more general as $\beta \in (0, \infty)$ can be any positive scalar, whereas $m’$ must be a positive integer.

---

> > ### Comment · Reviewer_pSUq · 2022-08-06
> > **I thank the authors for the detailed comment**
> >
> > I thank the authors for the detailed comment, and for clarifying the contributions of their work. I encourage the authors to revise the paper to emphasize those points.
> > I raised my score to 7.

---

### Official Review · Reviewer_9zyv · 2022-07-10

**Rating:** 5
**Confidence:** 3
**Soundness:** 3 good
**Presentation:** 3 good
**Contribution:** 2 fair

**Summary:**

The paper studies the PAC-Bayes generalization bound, which is an important topic and has been successfully applied to produce non-vacuous generalization bounds for neural network. The authors extend the existing results to the multiclass setting by introducing the discretized error types, which is a disjoint partition of the assembly of model prediction and ground truth labels. Based on the results, the authors further provide an abstract implementation of the method which can be applied to the real tasks.

**Questions:**

Although the discretized error types can be arbitrary in the theorem, are there any suggestions to set this parameter in the real tasks?

Can you make more detailed discussion about your results that whether they are vacuous or not?


**Strengths And Weaknesses:**

Strength:

1.The derived theorems hold for multiclass problem and soft labels, which has a wide range of applications to the real learning tasks. Such results in the PAC-Bayes literature have not been well-studied yet.

2.Although there are no empirical studies to validate the results, the authors provide abstract implementation of the algorithm for better understanding the applications of the theorems.

3.The paper is well presented and easy to understand. Detailed proofs are provided in the paper and supplementary material.

Weakness:

1.The technical contribution is somewhat weak in my view. The authors leverage the discretized error types to incorporate the entire confusion matrix, but many steps of the proofs mainly follow existing results.

2.For the minor problems, there are some flaws in the paper, e.g., incorrect citations in L13 in the paper.

---

> ### Author Response · Authors · 2022-08-01
> **Thank you for your time and effort in evaluating our paper. We are especially encouraged by your positive comments on the potential impact of our work, and its clarity.**
>
> You are correct in noting that the structure of our proof for Theorem 3 is very similar to, and indeed inspired by, existing proofs of other PAC-Bayes bounds. However, these methods only take one so far; the bound in Theorem 3 is far from usable in its current form. It takes considerable extra work to arrive at the truly useful bound given in Corollary 7. Indeed, the result rests upon the (to our knowledge) novel Lemma 8 (proved in Appendix B.2 of the supplementary material), which itself requires two helping lemmas. Irrespective of proof details, we believe that the novelty of the discretised error types approach makes the resulting bounds highly interesting, since they provide far richer measures of generalisation than hitherto obtained in the literature. We hope that this work is an initial step in opening up a new way of understanding generalisation, where one seeks not just to bound the expected performance, but to control the entire distribution of errors that an algorithm might make. This is an especially important question to address from the perspective of industry and, as reviewer x9gX points out, has ramifications for the fairness literature. We will revise our manuscript to better highlight the “wide range of applications” (reviewer 9zyv). Indeed, while we see our contributions as mostly theoretical, we believe the potential applications of our work are important and exciting, and the richer understanding of the distributions of errors for machine learning algorithms it enables may be quite impactful.
>
> To answer your first question, how to optimally discretise the error space is a highly problem-specific consideration, which is why we have deferred discussion of this point to a follow up (in-progress) companion paper which will explore practical implementations of the bound in much greater detail
>
> While we did not include experiments, mostly due to time and space constraints, we feel that our contributions are mostly theoretical in nature and significant in their own right, and that an extensive numerical study would be a distinct contribution. Nevertheless, two general comments can be made in response to your question on how to choose the user-specified error types. First, our preliminary numerical experiments indicate that the logarithmic term in the bound in Corollary 7 grows approximately linearly in $M$, so that $M$ should not be chosen too large. Second, this implies that one should only divide into separate types errors that one really wishes to distinguish in the given problem. We further note that our initial results indicate that the bounds are indeed non-vacuous on MNIST for reasonable choices of $M$, which we hope goes some way to answering your second question.

---

> > ### Comment · Reviewer_9zyv · 2022-08-09
> > **Thanks for the response**
> >
> > Thank the authors for the response. I partially agree with the claim of authors. Overall I still think this is a borderline paper with positive vote.

---

> > > ### Author Response · Authors · 2022-08-09
> > > **Request for details**
> > >
> > > Thank you for your response. Could you possibly elaborate on what you partially agree and disagree on so we have a chance to provide arguments?

---

> ### Comment · Area_Chair_wX61 · 2022-08-08
> **please acknowledge the authors' response**
>
> Please acknowledge the authors' response.

---

### Official Review · Reviewer_x9gX · 2022-07-10

**Rating:** 5
**Confidence:** 4
**Soundness:** 3 good
**Presentation:** 3 good
**Contribution:** 3 good

**Summary:**

This paper sets out to extend the PAC-Bayes framework to establish new generalization bounds for multi-class classification with M classes of errors extending the 0-1 risk. The main results appear to be Theorem 3 and Corollary 7, which generalize previously known results in this space, and can recover those existing results. My biggest concern about this paper is that I don't find the problem setup inspiring. In particular, the only worked out examples are in Section 4. The novel and interesting piece seems to be the definition of kl^-1, but I find the study to be thin. To sum everything up, I think the derived results appear to be correct; there is some novelty in proofs; but the setup of the problem is not well motivated and it is not clear how these results may be used to derive intuition or practical algorithms.

**Questions:**

* As one example I can think of: can you say anything about balancing false positive rate and false negative rate in a binary classification?

* In particular, can you probably make any connections with the fairness literature where the goal might be to equalize false positive rate and false negative rate?

**Limitations:**

As it currently stands, the impact of the paper is not well quantifiable.

**Strengths And Weaknesses:**

Strengths
* The paper is clearly written and the theoretical derivations are clearly discussed

* Theorem 3 extends the existing results in the literature, and the proof requires some nice extension of existing results including Lemma 5.

* Corollary 7 is also a nice extension of the existing results. While I did not follow its proof closely, the form of the result appears to be meaningful.

* The new results motivate some new study, including a new definition of kl^-1 that seems to be interesting.

Weaknesses:

* The setup of the problem is not well motivated, and the study has not resulted in deriving new intuitions.

* The new definition of kl^-1 is not well motivated and the details around it are thin.

---

> ### Author Response · Authors · 2022-08-01
> **We thank you for your time and effort in evaluating our paper and we are especially encouraged by your positive comments.**
>
> Thank you for your kind and encouraging comments on the positives of our paper.
>
> We appreciate that our work departs from the current literature and we are happy to have the opportunity to better highlight why we believe our setup is relevant and hopefully will inspire new directions in the statistical learning community and, more broadly, to ML practitioners. Hitherto, PAC-Bayes bounds have been restricted to bounding only one-dimensional measures of risk, for example, the probability of error or, more generally, the expected loss. We feel this is an impoverished view, and we will illustrate this in the simple case of binary classification. Suppose the task is to classify a medical image as containing either a benign tumour or a potentially malignant one requiring a biopsy. Then the balance of type one and type two errors must be finely controlled. It is true that ordinary PAC-Bayes can account for this by weighting the different error types and then bounding this one-dimensional metric, but the problem comes when the balance needs to be changed. Suppose, for instance, that it is later discovered that biopsies are riskier than expected. In this case, the ordinary PAC-Bayes bound is rendered useless and retraining is required to account for a different weighting. The only way to bound multiple weightings simultaneously would be to use a union bound, at the expense of a degraded (less tight) resulting bound. The significant advantage of our method therefore is that it bounds *all uncountably many weightings simultaneously*, something that could not be achieved with a classical PAC-Bayes bound, even with a union bound. We require neither a union bound nor retraining. A far richer measure of generalisation is thus provided – one obtains a bound on the type one error, a bound on the type two error and bounds on every possible weighting of the two. All of these bounds hold with high probability simultaneously. In our example, better estimates of the risks of biopsies could be seamlessly integrated without the need for retraining or a union bound. In multiclass classification, where it may be harder to estimate the costs of different errors, this flexibility may be especially valuable.
>
> We hope that helps elucidate our motivation to tackle this important problem. Further, we hope we have answered your pertinent questions on the connection to the fairness literature (see, for example, “Differential privacy and generalization: Sharper bounds with applications” Oneto et al. 2017, or “PAC-Bayes and Fairness: Risk and Fairness Bounds on Distribution Dependent Fair Priors.” Oneto et al. 2019), in particular balancing the rates of false positives and false negatives. While our contributions are distinct from these lines of work, we will revise the paper to better emphasise the connections should it be accepted.
>
> As to your comments on our extended definition of the kl inverse, if the analysis appears to be thin that is because we use it as a means to an end and it should not be interpreted as profound. In Section 4 we use the extension of the definition of the inverse kl to translate the bound on the kl given in Corollary 7 into bounds on weightings of the error types. We carefully derive the properties of the kl inverse that are required for this purpose – namely well-definedness and how to calculate its derivatives – in Appendix B.4 of the supplementary material. If the paper is accepted, we will better highlight that the extended definition is a technical tool created for this purpose.
>
> As for practical algorithms, we discuss this in detail in Appendix A of the supplementary material “Recipe for implementing and deploying our strategy.” While we did not include any experiments ourselves (mostly due to time and space constraints – although we also feel our contributions are mostly theoretical in nature and significant in their own right), we hope that the instructions provided demonstrate that the method is highly practical. Indeed, it follows very closely the usual approach to implementing PAC-Bayes bounds found and frequently used in the literature (for example, see “Computing nonvacuous generalization bounds for deep (stochastic) neural networks with many more parameters than training data”, Dziugaite and Roy, 2017). The only additional step in implementing our bound is the use of Proposition 11 to obtain derivatives for kl^-1, which, despite appearances, is straightforward to implement.
>
> We acknowledge that our contributions are theoretical and we believe that a thorough empirical analysis calls for a separate paper, which is a current work in progress that will be submitted to a different venue.

---

> > ### Author Response · Authors · 2022-08-09
> > **Reminder**
> >
> > Dear reviewer, we hope that you have had chance to read and consider our response to your review, and that you would be able to share your thoughts with us.

---

> > ### Comment · Reviewer_x9gX · 2022-08-09
> > **Practical impact yet to be demonstrated**
> >
> > Dear authors,
> >
> > Thanks for your response! While I like the binary example you give about unequal error interesting, I still don't see how this can be applied practically. Can you please turn this into a real experiment, and demonstrate that existing methods fall short, and quantify benefits of the new bound?
> >
> > Thanks for sharing references on connections between your work and fairness literature. I asked this question as a potential area for the authors to work out the bounds and perhaps show something new and non-trivial in an important practical problem.
> >
> > As it stands, I think the theory is correct; there is some novelty in how the proofs are extended from previous work, but I am yet to see a concrete application for the new results. As a pure theoretical paper, I think the contribution of this paper is borderline. I would be inclined to be supportive of acceptance if the authors can showcase how the extended theory can shed insights in practice. IMHO, the examples discussed in words are not sufficient to motivate the paper. I hope the authors can apply these new bounds to a real problem and actually show something new and insightful!
> >
> > Thanks, \
> > Reviewer x9gX

---

> > > ### Author Response · Authors · 2022-08-09
> > > **Experimental validation not necessary for our qualitatively new result.**
> > >
> > > Thank you for your response to our comment. We would hope that our example, along with the detailed explanation in the paper and the supplementary material of how to turn the bound into a training objective would be sufficient to convince readers of the practical utility of our method and to demonstrate what it can achieve over existing results. Of course at such short notice it is not possible to construct an experiment. But, since the setting and approach of our method is theoretically novel, it is not in fact comparable to existing methods. To stress once again, no existing bound can offer what ours does, namely uncountably many bounds (one for each weighting) simultaneously. No experiment is necessary to demonstrate this - it is evident from the theory.

---

### Author Response · Authors · 2022-08-01
**Warm thanks for your work and detailed comments!**

We warmly thank the three reviewers for their evaluation of our work, and we are pleased to read that they found the paper “clearly written” (reviewers x9gX and 9zyv), with novel and “nontrivial” (reviewer pSUq) bounds holding for multiclass classification and soft labels with a “wide range of applications” (reviewer 9zyv). We provide a point-by-point response to all reviewers below.

---

### Meta-Review · Area_Chair_wX61 · 2022-08-23

**Recommendation:** Reject
**Confidence:** Certain

**Metareview:**


The consensus was that the reviewers were not convinced that the results were significant, and did not see significant fundamental novelty in the analysis.


**Award:**

No

---

### Decision · Program_Chairs · 2022-09-14

Reject